# Microwave Resonators for Wearable Sensors Design: A Systematic Review

**DOI:** 10.3390/s23229103

**Published:** 2023-11-10

**Authors:** Iris Royo, Raúl Fernández-García, Ignacio Gil

**Affiliations:** Department of Electronic Engineering, Universitat Politècnica de Catalunya, 08222 Terrassa, Spain; raul.fernandez-garcia@upc.edu (R.F.-G.); ignasi.gil@upc.edu (I.G.)

**Keywords:** textile materials, flexible resonator, textile resonator, wearable device, washing analysis, bending analysis, embroidered resonator

## Abstract

The field of flexible electronics is undergoing an exponential evolution due to the demand of the industry for wearable devices, wireless communication devices and networks, healthcare sensing devices and the technology around the Internet of Things (IoT) framework. E-tex tiles are attracting attention from within the healthcare areas, amongst others, for providing the possibility of developing continuous patient monitoring solutions and customized devices to accommodate each patient’s specific needs. This review paper summarizes multiple approaches investigated in the literature for wearable/flexible resonators working as antenna-based systems, sensors and filters with special attention paid to the integration to flexible materials, especially textiles. This review manuscript provides a general overview of the flexible resonators’ advantages and drawbacks, materials, fabrication techniques and processes and applications. Finally, the main challenges and future prospects of wearable resonators are discussed.

## 1. Introduction

Over the past few years, the industry demand has driven research in the field of flexible electronics on textiles to enhance wearable sensing devices [1]. Resonators represent the key component for RF and microwave filters design on the medium- and high-frequency ranges (between 100 MHz and 1000 GHz) [2]. The first research toward microwave technology for filter development was published in 1937 by W.P. Mason and R.A. Skyes [3], where ABCD parameters were analyzed for filter derivation.

Decades later, the design of direct-coupled-resonator filters based on strip transmission lines intending to reach better responses at narrow bandwidths (and succeeding) appeared in the literature [4,5]. All the academical investigation propelled the improvement of filter manufacturing techniques, i.e., referring in the early 1970s to the suspended substrate stripline (SSS) approach over the traditional computer-aided designs to solve printed-circuits filters’ issues related to inhomogeneity in structures [6].

Resonators acquired a notorious significance within the sensing domain. The operational principle was based on the vibration at its natural resonance frequency as a response to the external excitations in the shape of return losses. Microwave devices’ circuits were fundamentally designed based on passive components—which manifested power dissipation and losses as long as the signal passed through—and active components operating in consequence to the passive component’s response to signal [7].

Over the past few decades, several studies found in the literature proved the efficiency of employing multiple coupled microstrip resonators to reach desired high selectivity and quality factor values [8]. The geometry design offer was again an advantage over other conventional sensing methods, meaning that microstrip resonators could easily adopt many shapes and configurations in order to grant the correct determination of the loss tangent and dielectric constant for the given sample either in sheet, liquid or paste [9], solid (i.e., meat) or gaseous (i.e., ammonia [10]) forms.

The main interest behind planar microwave ring resonators was the detection of permittivity variations—which were actually small in some cases i.e., foam [11]—in addition to the capability to capture in real time any change in the dielectric properties of materials. Heretofore, numerous successful investigations around microwave planar sensing technology were developed, those being specially oriented to the fields of biomedicine and physical medicine, the food industry, and aerospace, among others.

Nevertheless, the fundamental hindrance in the actuality is attaining the same outcomes when implementing this theory on textile substrates owing to the electrical and mechanical properties that, albeit advantageous from an industrial standpoint, might hinder the effectiveness of sensors since they are not electrically ideal materials. So far, studies manifested the effectivity of the embroidery technique to develop wearable antenna sensors, which gave the impression of being a great solution for wireless communication systems, i.e., GPS, WLAN, Wi-Fi, NFC, etc. in addition to emphasizing a series of advantages in terms of lightness, flexibility, mass production, comfort and bending capability [12]. Not only patch antennas but also many other topologies were found in the literature, where diverse resonant structures were selected to obtain the optimal capacitance at the time those were excited with a single port fed through the microstrip line. Recent examples of Ultra-High-Frequency Radio-Frequency Identification (UHF-RFID) antennas were built based on metamaterial (MTM)-based resonators [13] such as the open ring resonator (ORR) [12], symmetrical double-ring resonator (DRR) [14], split post-dielectric resonator (SPDR) [15,16], single and symmetrical double split-ring resonator (SRR) [17], and so forth. Along the lines of the antennas, the present work reviews what has been achieved to date in the microwave field in terms of approaching the microstrip ring resonator (MRR) geometry into rigid planar structures—aimed at material dielectric characterization—for its scalability in the development of wearable sensor devices on fabric substrates, which suggests an interesting investigation line.

This work is a review of the existing literature toward the usage of resonators for sensors design and the evolution of materials and techniques implementation for flexible electronics in the microwave spectrum. A wide-ranging exploration of numerous databases such as PubMed, ScienceDirect, Science Open, Nature, IEEE Xplore, Taylor & Francis, ACS Publications, Springer Link, Emerald Publishing Ltd., ACM, and IntechOpen was used to select conference proceedings and full-text articles. The following keywords were chosen: (wearable) and (flexible) and (textile or e-textile) and (resonator). From a final list of 280 results that were consulted, 154 of them were referenced in the article. Among these, 15% were state-of-the-art and sensing devices’ classification, 41% were theoretical studies about principle of operation, materials and techniques, and 44% were experimental analysis and applications.

The paper Is structured as follows. Section 2 briefly describes the principle of operation of microwave resonators, the techniques and the most common configurations. Section 3 enumerates and compares the conventional and new materials employed in flexible electronics for both substrates and conductive parts, focusing on their key properties according to the applications, and also outlines the existing manufacturing techniques, listing some examples. Section 4 is mainly focused on the theory of material characterization methods in order to further understand the major concepts to consider when selecting or designing the resonators for sensing purposes. Section 5 is a state-of-the-art of the flexible resonators and wearable textile resonator-based sensors, exposing the evolution of techniques and applications in multiple sectors. Finally, Section 6 concludes the study and suggests the future lines of investigation.

## 2. Principle of Operation and Classification of Microwave Resonators

### 2.1. Principle of Operation of Microwave Resonators

By definition, a microwave resonator is a structure based on a series of parallel RLC lumped or quasilumped-element equivalent circuits aimed at confining the electromagnetic energy. Microwave resonators are of interest due to the range of applications they offer, principally filters, oscillators, frequency meters and tuned amplifiers [18].

Ideally, a microwave resonator might be prototyped by a simple LC circuit configuration given a frequency close to the resonance [19], neglecting the losses.

According to the transmission line theory, in a case where the resonant circuit sizes are multiples of λ at the fundamental frequency (e.g., half-wavelength resonator, λ/2, or quarter-wavelength resonator, λ/4), the result is no longer acknowledged as a lumped element but rather a distributed line resonator.

The following subsections introduce the concepts of resonance frequency and quality factor to describe the sensing principle of microwave resonators.

#### 2.1.1. Resonance Frequency and Quality Factor: A Brief Description


**Resonance Frequency, *ω*_0_**


As mentioned in the previous section, in an RLC resonant circuit, the resonance occurs when the magnetic and electric impedances condition is met (*X_L_ = X_C_*). The resonance frequency, *ω*_0_, is expressed as shown in Equation (1):(1)$$i\omega L=\frac{i}{\omega C}\to \omega_{0}=\frac{1}{\sqrt{LC}}\;$$


**Quality Factor, Q**


The Q-factor measures the loss of the resonant circuit as the ratio of the resonance frequency to the bandwidth of the circuit, meaning that the higher the *Q* value, the lower the losses.

The losses in a resonator might have different origins. The *Q* isolated from any external circuitry effect (without the MUT—Material Under Test—being loaded) is designated as unloaded *Q*_0_. This is the higher Q value for the system.

Equation (2) describes the relation between the loaded *Q*, the resonance frequency *f*_0_ and the ±3 dB (half-power) bandwidth of the frequency, Δ*f*. The same is illustrated in Figure 1.
(2)$$Q=\frac{f_{0}}{f_{2}-f_{1}}=\frac{f_{0}}{\Delta f}$$

The unloaded *Q*_0_ has a logaritmic relation with the inserion loss *IL* (obtained from *S*_21_, measured in dB) as per Equation (3):(3)IL=20log(1−QQ0)→Q0=Q(1−10−IL20)

Therefore, the loaded *Q* bandwidth, *BW*, for a circuit with bandpass properties, is given in Equation (4):(4)$$BW=\frac{\omega_{0}}{Q}$$

The insertion loss increases for a *loaded Q* near to an *unloaded Q*_0_, and a higher *loaded Q* leads to a narrower bandwidth, which can be very lossy. Hence, in order to avoid excessive losses, *unloaded Q*_0_ >> *loaded Q.*

Since the Q-factor is decisive for the selection of a resonator design (for instance, for filtering applications), multiple configurations have been presented throughout the years in the literature. The length of the lossy transmission line and its termination are the two variables to adjust in order to achieve the desired results. Three typical configurations are enumerated in in Table 1.

#### 2.1.2. Frequency Variation as a Sensing Principle

The resonators of interest in this article sense the variation of the resonance frequency caused when placing the MUT in the sensing area. These are extensively used for the dielectric characterization of solids, liquids and even gases by studying the relation between the *f*_0_ and the loaded *Q* with the material’s dielectric constant or permittivity, *ε_r_*.

The resonator consists of a sensing area (where the MUT will be placed) and a transmission line. First, an initial value of the resonance frequency without the MUT *f*_0_ is obtained, and then the transmission line is loaded with the MUT, which causes a variation of the resonance frequency, *f*_0_’. The dielectric properties values are hence obtained by the frequency response shifting of the *S*_21_ parameter.

Figure 2 illustrates the frequency variation before and after placing the FUT.

### 2.2. Classification and Configurations of Resonators

#### 2.2.1. Types of Resonators

Resonators are classified into three main categories: transmission line resonators, waveguide cavity resonators and dielectric resonators. The interest of the present article is the transmission line resonator, specifically the microstrip resonator, which encompasses one or multiple oscillating electromagnetic fields. The transmission line resonant method is a better choice compared to others such as coaxial or waveguide cavity in terms of compactness in size, robustness, elevated sensitivity, and especially because it is easy to fabricate and does not imply the destruction of the MUT. All these benefits make this method the most viable for the design of wearable devices.

These can be classified as below.

○Lumped-element resonator and quasilumped-element resonator, which resonate accordingly to the aforementioned Equation (1).○Distributed line resonators, which resonate at a frequency, *f*, that is a function of the fundamental resonance frequency, *f*_0_.
Quarter-wavelength (*λ*/4) resonator (*f* ≈ (2*n* − 1) f_0_ for *n* = 2, 3, …) with shunt series or shunt parallel resonance.Half-wavelength (*λ*/2) resonator (*f* ≈ *nf*_0_ for *n* = 2, 3, …).Ring resonator, whose resonance frequency is a function of the ring radius, *r* (2π*r* = *λ*, *f* ≈ *nf*_0_ for *n* = 2, 3, …).And others: stepped impedance resonator (SIR), hairpin resonator, etc.○Patch resonators, which can be shaped into numerous geometries such as circular or triangular.

Figure 3 illustrates the above-mentioned resonators.

Distributed line resonators can be configured in several ways depending on the filtering requirements. In ring resonators, the symmetrical geometry permits the resonance to befall in any orthogonal coordinate, which is a feature of real interest for the design of dual-mode filters [19]. Alternatively, other dual-mode resonating structures have been designed and implemented in both wired and wireless communication methods, such as squared and mender-loop resonators, where this last one is advantageous due to the compact dimensions [20].

For sensing purposes, the typical topologies of resonators that are widely employed are listed below:Electric-LC (ELC);Ring resonators:
○Split-ring resonator (SRR);○Complementary split-ring resonator (CSRR);○Open split-ring resonator (OSRR);○Open complementary split-ring resonator (OCSRR);○Spiral resonator (SR);○Complementary spiral resonator (CSR).


Figure 4 illustrates these topologies and their equivalent circuits.

## 3. Materials and Methods in Flexible Electronics

Flexible electronics are considered those electronic devices with the capability to alter their planar structure by bending, folding, twisting, stretching and deforming without altering their electrical properties.

The first investigation toward flexible electronics dates back to the 1960s, when a flexible solar cell arrays prototype made of crystal silicon wafer cells assembled on a plastic substrate was presented [22].

From then onwards, a huge evolution can be observed in terms of design and materials selection in order to achieve the desirable performance, dimensions and adaptability of the electronic devices to match and supersede the huge market demand.

The materials must prove, at least, exceptional mechanical properties and energy generation and storage. Other promising properties might be water resistance, high adhesion, biodegradability and so forth. The author of [23] enumerates the six requirements of flexible substrates:Optical properties: low birefringence and clear substrates are a must for displays.Surface roughness: asperities impact on the electrical function.Thermal and thermomechanical properties: the substrate’s working temperature must support the maximum fabrication process temperature.Chemical properties: avoid contaminants release.Mechanical properties: an elevate elastic modulus is necessary to grant the substrate’s bending and stretching.Electrical and magnetic properties: substrate must either conduct or insulate as per the designs’ requirements.

Multiple layers have to be considered when creating flexible electronic devices: the substrate, the connectors/electrodes and the conductive/functional layers. In [24], a classification of intrinsically soft materials for flexible electronics is given. All details are reported in Table 2.

Section 3.1 summarizes the properties of the main conventional substrates employed in the design of flexible electronics. Section 3.2 focuses on e-textiles properties and manufacturing techniques.

### 3.1. Conventional Substrates

Traditionally, three different types of materials are used as substrates for flexible applications: metals, organic polymers and thin glass. Their properties are compared in Table 3.

### 3.2. E-Textiles

Given the increase in the manufacturing demand for wearable electronics for body area networks (BANs), the industry is working on the usage of textiles so these can be employed for the design of clothes, shoes and other accessories such as wristwatches. Textiles have been of strong interest in health since the 1990s for the constant monitoring of patient vitals and activities such as electrocardiography, electromyography and electroencephalography. Textile-based electrodes and sensors can be employed for sensing strain, pressure, temperature, heat flux, sweat, humidity, and even gas detection [27].

According to ISO-TR-23383-2020 [28], there is a differentiation between the electrically conductive textiles and the smart textiles (e-textiles): electrically conductive textiles are defined as textiles with integrated e-circuits made of conductive fibers, whereas e-textiles are textile-based products with electronics embedded on them.

The following subsections enumerate the main properties of the textiles for electrical applications and the most common textile substrates and conductive fibers and yarns, according to the literature.

#### 3.2.1. Key Properties of Textiles in Electronics

The huge range of fabrics and their particular properties offers the flexibility to select the best choice for each specific application and its desired performance. Several are found in the literature investigating the RF applications (mainly the design of antennas and resonators) on fabric substrates. One of the most commonly used materials is felt, but there are others like goch, leather, jeans, cotton, silk, wax or even combinations of those [29].

In order to not compromise the performance of the sensing device, it is crucial to study and select the appropriate textile. Depending on the industry demand and the applications, some intrinsic properties will be more instrumental than others, either enhancing the electrical capacity and/or granting mechanical features and quality.

In this section, we comprehensive review the key properties according to the literature: Relative permittivity (dielectric constant);Loss tangent;Relative humidity and regain (absorption of moisture);Surface resistivity, resistance and conductivity;Thickness;Flexibility and elasticity;Mechanical deformations.


**Relative Permittivity and Loss Tangent**


By definition, the relative permittivity, *ε_r_* (also known as ‘dielectric constant’, *k*) determines the ability to polarize a material when exposed to an electrical field [30]. In other words, this is an intrinsic material’s parameter that indicates its capacity to store energy when applying a potential across it [30].

The relative permittivity represents the real part of the complex dielectric permittivity, ε_*r*_, which is expressed as shown in Equation (5):(5)$$\varepsilon =\varepsilon_{0}\varepsilon_{r}=\varepsilon_{0}\left({\varepsilon^{\prime }}_{r}-j\varepsilon^{\prime \prime }{}_{r}\right)$$
where the permittivity of vacuum, $\varepsilon_{0}$, has a value of 8.854 × 10^−12^ F/m.

The loss tangent, *tanδ*, is the relation between both the imaginary and the real part, which quantifies the material losses. A small loss tangent (*tanδ* << 1) is found in low-loss dielectric materials, whereas a large loss tangent (*tanδ* >> 1) is inherited in lossy dielectrics that demonstrate a limited capacity to store energy [27].

The loss tangent is expressed in Equation (6):(6)$$\varepsilon =-\frac{\varepsilon^{\prime \prime }{}_{r}}{{\varepsilon^{\prime }}_{r}}$$

Hence, these two parameters are certainly decisive in the selection of a material for RF microwave applications.


**Relative Humidity and Regain**


Relative humidity, *RH*, expresses in percentage the ratio of a water–air mixture against the saturation humidity ratio at a specific temperature.

Moisture regain, *MR*, expresses the moisture absorption, in other words, the capability to absorb water or moisture from the environment, which is also a percentage. Figure 5 exposes the moisture regains of some natural fibers, according to [31]. Others of interest are found in [32,33].

Both relative humidity and temperature have a huge impact on the fibers’ physical properties such as dielectric constant, tensile strength, electrical resistance, rigidity or elastic recovery. The effects will be more notorious in natural and hydrophilic fibers than synthetic polymeric fibers, which are hydrophobic.

In [34], a breaking load test is performed on cotton, polyester and silk samples to compare the effect of the relative humidity on the tensile strength of the fabrics. The results prove that silk and cotton are highly conditioned by the moisture in the environment due to their hydrophilic nature, which results in an evident degradation of their fibers in the case of silk, whereas cotton tends to increase the strength the higher the *RH* is. This is explained by the cellulosic properties of the material, which permit the release of internal strain by swelling, resulting in a uniform distribution of stress.

In the case of polyester, due to its hydrophobic characteristics and crystalline structure, the tensile strength remains almost unaltered along the different relative humidities.

Relative humidity also has an influence on the electrical properties, in a sense that the presence of moisture alters the relative permittivity, *ε_r_*: the higher the *RH* is, the more the *ε_r_* increases (due to the bigger presence of water, which has a relative permittivity value around 78.4 [35]).

In [36], a study over five samples is performed to analyze the influence of the relative humidity in the electrical properties of fabrics for antenna design. A quadratic relation between the *RH* and the *ε_r_* is expressed in Equation (7), which explains how the relative permittivity increases more in accordance with the relative humidity on materials that are naturally more absorbent.
(7)$$\varepsilon_{r}=a{\left(RH\right)}^{2}+b$$

It is also observed in the aforesaid article that the frequency is shifted to lower frequencies as the relative permittivity increases, and the frequency peak widens and flattens as a consequence of additional losses.


**Surface Resistance, Surface Resistivity and Conductivity**


Surface resistance, *R_s_*, defines the capacity that a material has to resist the electricity flow. In other words, it is the relation between the DC voltage and the current that flows between two electrodes, according to Ohm’s law. It is measured in Ohms (Ω).

The surface resistivity, *ρ_s_*, is the resistance to current leakage along an insulating material’s surface. Its measurement is in Ohm/square (Ω/sq.) [37].

Electrical conductivity or conductance, *σ*, is defined as the electrical conduction, meaning the material’s ability to transfer current. The SI unit of conductivity is Siemens per meter (S/m) [38].

These three parameters are related in that the conductivity is inversely proportional to the surface resistivity per a given thickness, *t*, as shown in Equation (8).
(8)$$\sigma =\frac{1}{\rho_{s}\cdot t}$$

For this reason, it is so important when selecting the conductive fabric for sensing device manufacturing to consider also these three parameters that determine the electrical performance.


**Thickness**


The substrate thickness is another critical parameter to consider in the design of antennas, sensors, and so on. The fabric thickness and the number of layers, along with the relative permittivity, determine the bandwidth, the gain and the efficiency performance of the corresponding planar antenna or resonator.

Textile materials are generally limited to a narrow range of relative permittivity. Therefore, the fabric thickness becomes more decisive when trying to maximize or minimize the bandwidth. This is explained by the relation with quality factor, *Q*, according to Equation (9).
(9)BW~1Q
where the *Q*-factor results of the reciprocal relation of losses are shown in Equation (10): space wave (*Q_rad_*) losses, conduction ohmic (*Q_c_*) losses, surface waves (*Q_s_*) losses and dielectric (*Q_d_*) losses.
(10)$$\frac{1}{Q_{t}}=\frac{1}{Q_{rad}}+\frac{1}{Q_{c}}+\frac{1}{Q_{d}}+\frac{1}{Q_{sw}}$$

Therefore, the relation with the thickness of the substrate has to do with these losses individually: one characteristic of thin substrates is having elevated space wave losses (also known as ‘radiation losses’). This means that the thicker the fabric, the smaller the overall Q-factor (*Q_t_*). Consequently, a larger bandwidth is allowed for a thicker substrate [39].


**Flexibility, Elasticity and Mechanical Deformations**


Textiles are key given their excellent flexibility and elasticity that allows their perfect adjustment in curved surfaces, as they can bend, elongate and compress. However, these properties may also be a disadvantage, as they lead to inaccurate measurements and dimensions when designing resonators and other electrical components. Also, it is found in the literature that modifying the shape of the sample does also imply a change in the resonance frequency of the device.

Not only this but the fact of bending has also a direct impact on the electrical properties, particularly the bandwidth, since the substrate thickness becomes irregular. This is an important parameter to measure when intending to use the device for human body applications. Here, the weave structure plays an important role as it might enhance the tensile strength. A wider explanation is provided in Section 4.2.

#### 3.2.2. Textile Substrates

Several articles are found in the literature investigating the RF applications (mainly the design of antennas and resonators) on fabric substrates. One of the most commonly used materials is felt, but there are others like goch, leather, jeans, cotton, silk, wax or even a combination of those [29].

Textiles’ dielectric properties are fully dependent on the dielectric behavior of their fibers, which are generally constituted by polymers. This explains why the relative permittivity of fabrics is so low compared to conventional electronics-related materials in the view of the fact that polymers are largely treated as insulators due to their low electrical conductivity [40].

The dielectric properties of some typical fabrics are listed in Table 4.

#### 3.2.3. Conductive Fibers and Yarns

The factors that determine the performance of fabrics are, principally, the nature of the fibers, the yarn, the structure of the weave, the density of the thread, the air volume and the surface treatment, if any [45]. However, fabrics are susceptible to environmental conditions such as pressure, humidity, and temperature.

Fabrics are naturally insulating materials made of fibers. It is possible, though, to combine them with metallic fibers, filaments or coatings, or even with conductive polymers, so they become what is known as conductive fabrics or electro textiles. The first reference to this concept found in the literature was in 1952 by S.P. Hersh [46].

Dielectric fabrics are characterized for their really small or negligible conductivity. These are used in electronics as the dielectric substrate for the design of wearable and flexible devices, e.g., sensors, antennas or resonators. Here, the dielectric properties of the textile do truly determine their gain and efficiency.


**Conductive Fibers**


Conductive fibers can have a metallic, carbonic or polymeric nature. Their properties are enumerated in Table 5.


**Conductive Yarns**


Conductive yarns are manufactured using purely metallic wires or they are simply core-covered by metal (usually stainless steel) having a conductivity range from 5 Ω/m to more than a few kΩ/m [50]. The classification found in the literature [51] is listed below.

Metallic filament yarns (metallic or metallized yarns);Conductive hybrid yarns;
Conductive composite yarns or metal composite embroidery yarns (MCEYs);
▪Core spun yarns;▪Commingled yarns;▪Blended spun yarns;Bi/multi-component spun yarns;
Inherently conductive polymer yarns (ICP);Laminated/coated conductive yarns;New conductive material and nanomaterial yarns.

The selection of the conductive yarn depends on the desired application, provided that each material has a specific conductivity level. Table 6 summarizes the possible applications for each yarn category.

### 3.3. Manufacturing Techniques

In this section, a comprehensive review of the most commonly employed techniques for manufacturing flexible electronics is given. Section 3.3.8 compares all the procedures and provides a set of applications found in the literature for each of them.

#### 3.3.1. Etching

Etching is a chemical process whose basic principle is to engrave the desired pattern into a substrate by causing a corrosive effect between the photoresist masks and the etchants. In other words, the materials are removed from the board with liquid chemicals. This technique began in the 1960s, and it is also known as chemical etching or liquid etching [53]. Etching allows the manufacture of complex patterms with a high level of accuracy compared to other technologies.

#### 3.3.2. Soldering and Welding

Both soldering and welding are joining techniques. These are widely used in flexible textile electronics to connect components and connectors to the conductive textile substrates (threads, flexible copper wires, polymides) [54].

In soldering, a filler metal (or solder) is melted to join two contacts. This third metal must have a lower melting temperature than the conductive thread or fabric. The conventional soldering technique requires high temperatures (above 200 °C), whereas lower temperatures (150 °C) are enough and required when working with fabric. However, not all conductive threads are suitable for soldering, for instance, stainless steel. The solder is a soft alloy of lead (Pb), tin (Sn) or silver (Ag) [55]. Some examples of the soldering technique applied to smart textiles is the integration of LEDs to the textile substrate using soldering conductive yarns [56].

Welding, similar to soldering, requires high temperatures to melt in this case both the metal and the textiles so they can be joined. This technique is seldom employed for e-textiles manufacture given the temperature conditions that limit the appropriate materials to use. The three main categories [57] are outlined below:Resistance welding: This requires two metallic parts to be joined by applying an electrical current through them.Ultrasonic welding: The ultrasonic acoustic vibrations with frequencies between 20 and 40 kHz generate heat and pressure that end up welding the two parts [58]. This is the most popular welding process with applications in multiple areas such as automotive, aerospace and health. An example is [59], where rotating ultrasonic welding technology is employed to integrate highly conductive ribbons into textile-based conductive tracks.Radio-frequency (RF) welding: The RF absorbing polymer tape is positioned in the middle of the two fabric pieces, which melts by submitting to RF energy. This technique is of interest for non-weldable fabric materials.

#### 3.3.3. Adhesive Conductive Foil

Conductive adhesive is commonly used in the shape of a sheet or strip especially for prototyping due to its low cost and easiness of employment. Adhesive copper tapes can be stuck or glued to many surfaces, e.g., ceramic cylinders [60], copper-clad substrates [61] or even textiles [29,62].

Copper foil tapes are commercialized in rolls and sheets of many dimensions and thicknesses. It is a remarkably modest material provided that the manufacturing process can be as elementary as cutting the strips manually with scissors to drawing the layout using a dedicated software and after printing and cutting it with a draft cutter.

The results, though, may not be indicated for purposes others than protoyping and pretesting or academic when it comes to employing this technique onto textiles because of their low reliability due to a lack of adherence over time (which impacts the sensing accuracy) compared to other technologies such as printing or embroidery.

#### 3.3.4. Printing Technologies

A wide range of technologies can be employed for printing electronic components. These are advantageous for their lightness, low profile and low cost compared to the conventional technologies, e.g., etching [63].

The printing processes can be divided in two categories [64]:
Conventional printing: These are printing technologies that require a master (printing plate), which is basically the tool that enables the ink transfer to the printing substrate. The conventional printing processes are outlined below:Screen printing;Letterpress/flexography;Lithography/offset;Gravure printing.Non-impact printing: These are masterless printing technologies (also known as ‘non-contact printing’ processes). These technologies employ laser to transmit the information to the intermediate carrier that will afterwards transfer it to the final substrate. The non-impact printing processes are outlined below:Electrophotography;Inkjet printing;Ionography;Magnetography;Thermography;Photography.

The application will determine in the vast majority of cases which is the most suitable printing technique. Generally, the minimum dot (feature) size and the throughput are the two main parameters to check against. However, others may also be taken into consideration [65].

Out of all the list enumerated above, screen-printing and inkjet-printing technologies have become the most relevant processes in the design of sensors and other electronic devices.


**Screen Printing**


Screen printing consists of the usage of a squeegee to apply pressure to the ink paste over a screen stencil, which is placed on top of the substrate. The stencil acts as a mask, meaning that the ink is imprinted through the non-covered cavities.

This technology is widely employed in electronics due to the good compatibility of materials, the low cost and the easiness of manufacturing [61]. However, the need to use a mask introduced an additional expense and material waste that may not be convenient when rapid prototyping [66,67].

Many applications are found in the literature for the design of RF antennas [68], RFID [69], smart packaging [70], plant biomarkers measurement [71], nitrate detection in water [72], monitoring metal pollutants [73], other smart wearables [74], IoT applications [75] and even electroluminescent matrix displays [76].


**Inkjet Printing**


Inkjet-printing technology is a contactless alternative widely employed for R&D and some other specific applications which involve the fabrication of latches [64]. It can either work with liquid or hot-melt ink. The functioning consists of the deflection of charged ink droplets by electric fields, which draws the desired image controlled electronically [63].

This technology has many variants—continuous ink jet, thermal (bubble) ink jet, piezo ink jet—that are used in accordance with the drop diameter and frequency.

In the literature, many applications are found for printed electronics, such as printed circuits, transistors, and amplifiers, in the design of antennas, sensors, RFID and audio devices. Many flexible color displays have also been developed so far using this technology [64]. Also, in the field of microwave technology, a recent (2020) study [77] has proven the feasibility of integrating circuits into microfluidic systems for heating and sensing purposes.


**Comparing Screen Printing and Inkjet Printing**


Screen- and inkjet-printing techniques are the most effective options in the field of printed electronics. While screen printing requires a stencil or master, it is still the best choice for mass production given the price–scalability compromise. On the other hand, inkjet printing is more recommended for prototyping and testing as long as it does not require a master [65].

In terms of the quality of the textile print, an experimental study [78] discloses the pigmentation outcomes of each technique on non-treated and pre-treated cotton fabrics.

The results show that inkjet-printed samples have an ink density three times lower than screen-printing ones, meaning that multiple passes are needed to obtain the same. Despite that, the color saturation is not affected by this factor but rather by the fabric properties, returning higher values for pre-treated cotton samples in general but still lower when using inkjet ink due to their viscosity and surface tension.

Between the four primary ink colors (magenta, cyan, yellow, black), many differences are also found that relate the viscosity with the density on each fabric.

#### 3.3.5. Three-Dimensional (3D) Printing

With a wide range of commercially accessible printing methods and materials, additive 3D printing techniques have recently gained appeal, especially for the development of flexible antennas, mainly because of the wide range of materials that can be employed under the classification of polymers and resins, metals and ceramics.

Additive Manufacturing (AM) by 3D printing is attracting the attention of the scientific collective given the flexibility, cost-effective, simple and fast prototyping alternatives that it brings. The basic principle of this technology is the addition of subsequent layers of materials in the desired digital design until reaching the desired tridimensional figure. In this way, the width of the 3D-printed substrate will depend directly on the material used and on the number of stacked layers. Multiple AM techniques exist; the more common are listed below [79,80]:Fused Filament Fabrication (FFF) or Fused Deposition Modeling (FDM);Stereolithography (SLS);Digital Light Processing (DLP);Material Jetting (MJ).

Fused Deposition Modeling (FDM) is the most popular technique, which encompasses the high-temperature extrusion of the filament modeling the object from bottom to top. In the literature, several studies provide successful results of implementing FFF technology for the design of RFID antennas [81,82,83], broadband 5G on-package antennas [84], SRRs [85,86], microstrip resonators [87], dielectric resonator reflectarrays [88], and others.

#### 3.3.6. Embroidery

Embroidery has become one of the most commonly used manufacturing techniques in the field of smart textiles, as it has been extended to multiple applications in aerospace engineering, construction, automotive, medicine and biomedical, amongst others [89]. This technique comprises the application of conductive yarn materials in a patterned design into a textile substrate.

Standard embroidery (or two-thread system) is the regular system generally employed to apply onto the textile a combined fine metal wire along with the yarns [90]. The Tailored Fiber Placement (TFP, also known as the ‘three-thread system’) dates back to the 1990s [91] and is an automated process that allows total control over the orientation and the directionality of the fibers in a composite preform. Using this technique, it is possible to build tridimensional structures of many shapes. The TFP method is recommended for glass or carbon fiber threads which are rigid and thick [92].

#### 3.3.7. Weaving and Knitting

Weaving and knitting are two textile fabrication methods which allow the fabrication of 2D and 3D fabric structures by working with multiple warp and weft layers. The knitting process comprises the positioning of the yarns on a knitting needle defining consecutive loops or stitches that are connected either horizontally (weft) or vertically (warp) [93]. In the weaving process, the yarns are interlaced perpendicularly, which simplifies the exercise to insert the conductive yarn.

Weaving and knitting are recommended techniques especially when the desire textile surface’s dimensions are huge. The most remarkable properties of woven and knitted fabrics are outlined below:Lightweight;Portability;Skin comfort;Durability;Deformation resistance (woven fabrics);Elasticity and stretchability, air permeability, thermal retention (knitted fabrics).

#### 3.3.8. Comparison of Wearable Electronics Integration Techniques

Table 7 summarizes all the aforementioned techniques. Figure 6 illustrates some application cases of resonator design employing them.

## 4. Materials Characterization and Design of Resonators

Resonators are widely used for the characterization of the dielectric properties of materials. In this section, the dielectric measurement methods are enumerated first; then, the criteria to select the optimal method and the measurement process are described. Finally, a review of the main challenges and considerations when designing resonators is given.

### 4.1. Materials Characterization

#### 4.1.1. Dielectric Measurement Techniques

The characterization of materials’ dielectric properties is elementary in RF and microwave engineering, inasmuch as these explain the interactions materials have with electromagnetic waves. The information behind these properties is decisive for the design, process and control of any product [98]. For the time being, it is not yet possible to use the same resonant method to measure the complex permittivity, *ε_r_*, and complex permeability, *µ_r_*, of any material throughout the entire frequency spectrum. Instead, different techniques must be employed depending on the material electrical and magnetic nature, the frequency and the desired accuracy [99].

As a matter of course, the dielectric measurement techniques for materials characterization are classified based on the resonance, the transmission type and the structures [100].


**Non-Resonant Methods**


Non-resonant techniques are used when measuring across a wide frequency range. Employing non-resonant methods, the materials are characterized from either the reflection or the transmission/reflection method [101,102,103].

Non-resonant methods are outlined below:Coaxial probe or open-ended coaxial line;Transmission/reflection line (waveguide, planar transmission line, coaxial line);Free space;Parallel plate.


**Resonant Methods**


On the other hand, resonant methods are peculiarly highly accurate and are used to measure the properties of low loss materials that operate at microwave frequencies. This means that with this method, it is possible to measure very small MUTs, but the technique is limited to a narrow frequency band. Contrarily to non-resonant methods, this technique allows the material characterization for a few discrete frequency points.

Resonant methods are outlined below [103]:Resonant cavity;Open-ended resonant lines;Planar transmission line loaded resonators.

All in all, resonant methods are recommended when working with low and medium frequency, whereas a non-resonant method will provide more accurate results for high-frequency assays.

All the details of the above-mentioned methods are described in Table 8.

#### 4.1.2. Selection of the Dielectric Measurement Method

According to the literature [105], the following factors are highly recommended to be assessed in order to select the proper measurement method:Material category or nature;Destructive/non-destructive;Frequency band;Measurement required accuracy;Dielectric loss;Frequency;Contacting/non-contacting;Measurement cost.

Table 9 classifies the dielectric measurement methods based on the material nature and the sample preparation.

#### 4.1.3. Measurement Process: Deviation of the Dielectric Property

The material characterization process consists of an iterative flow based on the S-parameters measurement in the experiment against the previously obtained results in the simulation for the designed resonant model. The deviation process flow is illustrated in Figure 7.

### 4.2. Ring Resonators in Flexible Electronics: Design Considerations

Ring resonator configurations—such as the split-ring resonator (SRR) and complementary split-ring resonator (CSRR)—are the most common ones in the field of flexible electronics according to the literature. The reason is that these are simple structures which can be easily implemented, and they present low radiation loss compared to others.


**Coupling Method**


Microstrip ring resonators are, in a few words, transmission lines shaped in a closed loop. The coupling gap is the distance between the feed lines and the ring, which has a capacitive effect in the resonator’s circuit, directly affecting the performance of the resonator by modifying the field perturbation and consequently the losses.

The geometry of the ring resonator geometry has a direct effect on the losses and henceforth the Q-factor. This depends not only on the dimensions of the ring but the layout of the waveguide/feed lines and the above-mentioned coupling gap. Such variables can be easily parametrized for simulation purposes; therefore, the coupling method is a simple and low-cost technique widely employed in the ring resonator design according to the literature.

The main four conventional coupling methods [106] are described below and illustrated in Figure 8.

Loose coupling or matched loose coupling: There is a large distance between the feed lines and the resonator; hence, the coupling gap effect is negligible. It is characterized by high insertion and return losses (*S*_21_, *S*_11_) and a high Q-factor.The main advantage is the simplicity of its implementation.Enhanced coupling or quasi-linear coupling: The feed lines are entrenched to the ring structure to improve the coupling efficiency. This geometry offers low insertion and return losses despite the degraded Q-factor. This is the most appropriate option performance-wise.Line-to-ring coupling: A peripheric enhanced coupling alternative is presented to reduce the insertion loss and increase the coupling strength.Matched-stubs coupling: The matched stubs are connected to the ring, and the gap is between these and the feed lines. This results in a slight increase in the insertion loss but a relevant effect in the resonance frequency.The Q-factor remains constant regardless of the stubs’ dimensions variations as well as the coupling. Therefore, this is the least recommended technique.

The resonance frequency is determined as a function of the gap size that varies depending on the model geometry as well as the resonator dimensions. The best way to obtain the optimal resonator geometry is to simulate the possible combinations of ring and feed line dimensions along with the different coupling gap distances.


**Bending**


In addition to the dielectric properties, the physical conditions must be well evaluated when selecting the suitable substrate for the design of the flexible device.

Tolerance to mechanical deformations such as bending, wrapping, folding or twisting is one of the main challenges in the design of flexible devices not only in regard to the material’s limitations but also to the manufacturing technique. For instance, soldered contacts require a reinforcing, and embroidered and sewed designs must ensure the threads are strong and flexible enough to endure the bending stresses.

In [107], an edge-coupled split-ring resonator (EC-SRR) unit cell is modeled to reproduce the effects of bending over a virtual vacuum cylinder with a variable radius. The simulation results prove that there is an inverse proportionality between the curvature radii and the effective permittivity, since the normal plane wave incidence becomes oblique with respect to the surface curvature due to bent EC-SRR. In contrast, an increase in the aforementioned bending extent results in a minor increase in the effective permeability.

Looking forward at the further applications, ref. [108] proposes a highly sensitive microwave CSSR-based sensor for liquid characterization bent into a cylinder again of numerous radii, as illustrated in Figure 9. The results of the simulation and the experimental analysis conclude that the bending effect has a direct impact on the *S*_21_ response and frequency in a way that the frequency band is notoriously widened for a lower radius in the case of 0% concentration of water, and it also increases the frequency range between minimum and maximum liquid concentrations for a smaller bending radius. In terms of sensitivity, an increasing linear tendency of the Q-factor is drawn, meaning that the more the cell is bent, the lower the rate of energy loss and the sharper and narrower the peaks are; therefore, it can be assumed that it will entail a higher resolution for the shifting.

Ultimately, it is crucial to consider the bending loss when designing ring resonator-based structure to meet the performance requirements. It is demonstrated that the Q-factor is improved by increasing the ring radius. In addition to this, the waveguide might also be wrapped concentrically to the ring to counterbalance the bending loss [109].


**Washing**


The possibility of wearing clothes that integrate electronics opens up the necessity to make them washable without impacting their performance. Scarcely any article is found in the literature about the effect of the washing process on the effectiveness of the resonator.

Antennas and resonators must tolerate periodic laundry cycles. Their characteristics are measured in terms of reflection coefficient (*S*_11_) and radiation efficiency (*η*), which are compared against the washing cycles. Table 10 enumerates the results obtained in various articles.

One challenge of e-textiles is the mechanical deformations that they suffer when exposed to multiple washing cycles and contact with the chemicals of the detergents. The aforementioned experimental analyses conclude that the overall performance stays stable and insignificantly affected after drying the samples. However, the considerations listed below are decisive when considering long-term usage:Washing nature: hand-washing, laundry machine.Laundry cycles: number of cycles, duration, speed.Drying cycles: technique (air drying, ironing, etc.), temperature, pressure.Detergent nature: liquid detergent has a lower impact in terms of a fabric’s structural stability over the powder detergent.

It is also well observed that the physical aspect of the samples is altered as per the effects of the moisture, air and detergent: the oxidation of the silver and copper creates a color change in the conductor. Moreover, the repetitive washing cycles may result in a degradation and delamination of the printed conductive ink as well as the detachment of the connectors. This effect degrades the frequency response of the corresponding wearable resonators. In order to prevent it, the models are layered with a protective coating.

In terms of shielding effectiveness, *SE*, the investigations settle an approximative direct relation between this parameter and the number of washing cycles, which is tested at different levels of frequency [113]. In article [114], the effects of the temperature and washing on the electrical properties of conductive yarns are studied, and the results of the investigation demonstrate that the electrical resistance increases to some extent after each wash. In addition, the high temperatures lead to a fabric contraction resulting in a decrease in the conductivity. In order to not compromise the electrical conductivity of the device, it is crucial to control both the temperature and the washing cycles. Since no protocols exist at the moment for e-textile products, the only reliable option is to simulate and predict the effect of the water, temperature and detergent on the textile and electronic components. Article [115] proposes a protocol to forecast the damage of the e-textile based on two parallel prediction tests: washing cycle tests to count the washing stresses at each iteration and laboratory tests to study the equivalent stresses. The correlation of the abovementioned tests provides an estimated reliability for the usage of the textile.

## 5. Recent Developments of Microwave Technology in Flexible Electronics

In this section, a summary of the multiple applications of flexible electronics is provided with special attention paid to the printing techniques previously explained due to the increasing industry demand and recent scientific investigations.

Subsequently, a state-of-the-art review of metamaterials—as an innovative strategy to accommodate the industry demand—and embroidery and printing techniques—as key methods in the manufacture of such geometries—are given.

Finally, this section concludes enumerating the primary challenges in the design of resonators for the design of wearable devices.

### 5.1. Flexible Electronics: A Brief Summary of the Applications

Resonators have acquired a huge interest for wearable applications using textiles. According to the literature [115,116], wearable e-textile applications areas can be classified as shown in Figure 10.


**Healthcare**


According to the literature, one of the first industries where e-textile integration began was the medical sector. These devices are classified for use in diagnostic and monitoring as well as therapeutic areas; for example, some of the most recent articles published so far are focused on the scientific study of glucose levels for diabetes disease monitoring [117,118].

Diagnostic and monitoring devices include all those instruments used for monitoring vital signs but also others such as electrocardiograms (ECGs) or fiber-optic sensors (FOBs). Some are enumerated below [115]:-Vital signs monitoring devices;-Glucose monitoring devices;-Sleep monitoring devices;-Fetal monitoring and obstetric devices;-Neuromonitoring devices;-Body movement monitoring devices.

These sensors are typically integrated with undergarments, shirts, gloves or wrist bands, and the data are real-time measured and transmitted to external devices or databases.

Therapeutic devices employing wearable e-textiles are intended to replace traditional methods, including the following [115]:-Rehabilitation devices;-Respiratory therapy devices;-Pain management devices: facial and migraine pain treatment, neuromodulation devices for transcutaneous electrical nerve stimulation (TENS) and spinal cord stimulation (SCS) treatments, radiofrequency ablation and cryoablation treatments;-Insulin pumps, intrathecal infusion pumps and external infusion pumps;-Message therapy: thermal therapy and ultrasound or infrared therapy;-Others such as cancer and neuropathic pain treatment devices, lung ventilation treatment, photodynamic therapy, metabolic disorder investigations, phototherapy, etc.

Since wearable sensors are highly designed for industrial, scientific and medical purposes, many wearable sensors are meant to operate in the ISM band [119].


**Sports and Leisure**


Some of the aforementioned medical devices have generated interest from a commercial standpoint, especially for fitness solutions. The objective behind this is being able to monitor main parameters such as body temperature, pressure or movement for professional athletes but also for amateur sport practitioners. Given society’s accelerated rhythm of life, there is a need to provide non-invasive, wearable and flexible sensors that can be worn as a complement when practicing a sport or any other daily activity.

The main devices manufactured to date for this purpose are oriented to body temperature, pressure and movement monitoring. These textile-based sensors offer numerous advantages including flexibility and conformability for the user.

Temperature sensors are classified based on their principle use as resistance temperature detectors (RTDs), infrared temperature sensors and thermocouples. The main substrates are natural materials like pectin or silk, cellulose materials (paper), silicon gels like Ecoflex or PDMS, or polymers such as polyurethane (PU) or polyimide (PI). Particularly for textiles, temperature sensors are enumerated as follows [120]:-Thermoelectric-powered textile-based temperature sensors: the thermoelectric (TE) components that are printed onto the textile convert the temperature variation into electrical voltage.-Textile-based resistive temperature sensors: the printed conductive material changes by effects of temperature.-Fiber-optic temperature sensors: optical fibers’ properties are altered by means of thermal gradient change.

Smart textiles or e-textiles with pressure sensors are also of interest in the sports and leisure industry. Some companies have already commercialized smart socks to prevent patient falls by measuring the interface pressures exerted on bed-ridden and mobility-impaired patients and real-time alarming when there is a fall-risk patient standing or attempting to walk unassisted [29,121]. Pressure sensors composition is decisive for its integration into garments. In [30], three approaches are presented. On one side, the pressure sensors are made of an assembly of two main parts: first, the pressure-sensitive conductive fabrics, and second, the conductive fabric with the electrodes on conventional polymeric (PI, PET) films. In this case, the sensor has non-fabric components, restricting the human motion and downgrading the skin comfort. Alternatively, these sensors can be manufactured fully textile-based; however, the drawback is the weak adhesion of the assembled components. A possible solution found in the literature is to use an elastic piezoresistive textile layer and screen printing a set of flexible interdigital electrodes on a polyester textile substrate.


**Fashion and Entertainment**


In the fashion and entertainment industries, the focus is on the usage of flexible displays and screens to create communicative clothing. This new concept of intelligent apparel is captivating the society, and the demand is increasing. The usages of such clothing in fashion and entertainment are, mainly, for aesthetic personalization and network games.

The challenges encountered so far when designing them are the control interfaces and sensors’ positioning, the hardware required in order to store and process the data, and the weight and volume compromise of the batteries in order to achieve a sufficient energy autonomy [122].

Wearable displays offer the possibility to flex and deform in order to adapt to the human body without deforming their projections. Wearable displays are classified as Light-Emitting Displays (LEDs): OLED, PLED, PLEC, ACEL; and colorimetric displays (electrochromic, electro thermochromic) [32].

DC-driven LEDs consist of a sandwich of functional layers containing PLECs, OLEDs, phOLEDs and PLEDs, amongst others. Due to the high complexity of manufacturing, these are not good candidates for implementation onto textiles.

AC-driven LEDs, instead, are much easier to manufacture, since these do not require the assembly of layers: these can be set up into coaxial fibers or multilayered fabrics. Unfortunately, the AC-driven LEDs work under low values of luminance and high operating voltage.

Colorimetric displays are, as well, manufactured under the sandwich concept. The principle of operation of the electrochromic materials is the redox reactions that take place when applying an electrical potential across the electrodes in between which gel electrolytes are sandwiched. For the thermochromic materials, it is the Joule heating effect that triggers the color change when heating over what is called the activation temperature.

#### SAR and Bending Reliability Tests of Flexible Resonator-Based Sensors


**Bending Reliability Tests**


Flexible electronics’ most obvious advantage is the mechanical possibility to deform, flex, bend, twist and adapt to curve geometries—all this without impacting their electronic parameters. However, a critical aspect is that repeatedly bending might lead to electronic failures [123]. For this reason, the device reliability must be investigated.

In [124], an overview of the possible bending reliability tests is presented. In terms of design, it is suggested to avoid placing electronic rigid components on the high-strain areas. Three mechanical properties must be evaluated when applying a tensile stress to these components: breaking strength, breaking displacement and breaking stress. In order to do so, two different uniaxial bending tests are suggested: a three-point bending test and four-point bending test. A three-point bending test is recommended for rigid and semi-flexible components, whereas a four-point bending test is a better option for multi-flex layers. Others such as push and roll to flex are suitable measurements when higher displacements with very small radii are expected.

Several examples of analytical and experimental bending analysis performed in antennas are found in the literature. In [125], a bending study on an ultra-low profile rectangular patch antenna for wearable communication devices is performed: the simulated results are consistent with the experimental measurements, and a change in the transmission performance of the bended antenna is observed, which is explained by the mechanical vibrations and other disturbances.

Similarly, a review of the bending analysis of polymer-based flexible patch antennas is made in [126]. Polymers such as PDMS, PI, PET, PTFE and Kapton polyimide are considered as part of the study given the high industry demand of using such materials for flexible electronics design especially for IoT and wireless applications. The purpose of the article is to provide a detailed assessment of the mechanical and electrical properties of the aforementioned polymers and the impact on their performance when bending or deforming. It is concluded that the physical structure of the polymer is crucial, since the substrate height and the material’s thickness and width are the three key parameters to compromise, along with the feeding technique (to guarantee the impedance matching of the feeder with the patch antenna), and that some polymers have a better response to bending than others in a specific frequency range in terms of reflection coefficient and signal degradation.


**Specific Absorption Rate (SAR)**


When designing resonators and antennas for on-body applications, the specific absorption rate (SAR) is one of the most important measurements to consider, since it determines the amount of RF energy absorbed by the human tissue in terms of electrical conductivity, *σ* (S/m), per mass density of the tissue, *ρ* (kg/m^3^). SAR is determined as the EM power absorbed per unit mass, which is averaged over a certain amount of tissue. There are two maximal SAR values for the general public (head and trunk) and averaging levels depending on the country regulation. On the one hand, the 1.6 W/kg averaged over 1 g tissue is based on dosimetric considerations, and it is considered in the USA, Canada and Korea. On the other hand, the 2 W/kg, averaged over 10 g tissue, based on biophysical criteria, is applied in the EU, Japan and China. For occupational exposure, those values are increased up to 8 W/kg (1 g tissue) and 10 W/kg (10 g tissue).

The reason behind this is to ensure always the human body safety from the radiation exposure. Therefore, the objective is always to design wearable devices that certify a low SAR according to the international regulations and standards [127,128]. In [129], an exhaustive review of the SAR of multiple distinct topologies of textile antennas is presented with two main objectives: first, to provide a full analysis of the external factors that influence the SAR of an antenna based on simulations, and second, to prove the accuracy of the simulations against the measurements in actual samples. The results of simulations in the Hugo human model reveal that there is a direct dependency of the SAR value and the tissue and textiles conductivities, exposing a saturation over specific conductivity values in the numerical analysis. Moreover, is it also well proven that the higher the distance of the antenna-to-phantom is, the lower the SAR rate, reaching almost zero when at 20 mm. Finally, this article exposes the main challenges of measurement of SAR rate experimentally, which are mainly attributed to the positioning of the probe and the external conditions such as temperature and humidity, which alter the results of the estimated ideal conditions in computer simulations.

### 5.2. A Comprehensive Review of Printed Ring Resonators for Wearable Applications

Multiple printing technologies exist—as detailed in Section 3.3.4 and Section 3.3.5 in this article—for the design and manufacture of wearable devices. Printing offers a wide range of options for flexible substrate materials beyond textiles: polymers (including silicones, thin films, commercial specific polymers), thin and/or ultra-thin glass, and even paper-based sheets. One example is [130], where a ring resonator is printed on a flexible Parylene-N substrate for its characterization (refer to Figure 11a).

Hence, the applications are numerous and present many advantages compared to the other conventional manufacturing techniques described in Section 3.2. Even though ring resonators are simple structures geometrically speaking, these cannot be precisely embroidered due to the complexity of the circular form. For this reason, printing technologies are the best candidate to manufacture such shapes.

#### Inkjet-Printed Arrays

Inkjet technology is optimal for the mass production and the manufacture of arrays made up of symmetrically distributed unit cells. The most common structure is the split-ring resonator (SRR). In [131], an inkjet-printed SRR metamaterials array is presented as a low-cost solution intended to operate at the lower frequencies of the X-band potentially for frequency-selective microwave-shielding applications. The design consists of an arrangement of 3 × 20 × 15 SRRs uniformly and periodically distributed. In [132], a set of flexible electrically tunable metadevices based on reconfigurable microwave resonators is presented. A double broadside-coupled SRR model is proposed as an alternative to the conventional SRR structures with the objective of lowering the resonant frequency and form factor. In Figure 11a, an inkjet-printed array of SRRs is fabricated on a commercial polyimide substrate using a commercial silver nanoparticle dispersion and conductive polymer inks. Similarly, in [133], a flexible metamaterial PI film is fully inkjet printed with an array of silver rectangle-shaped SRR resonators with the objective of reaching high electromagnetic wave absorptivity. In the article, a parametric study is performed to obtain the dependence of the absorptivity on frequency at specific geometrical dimensions of the model. Three different sizes are used to print the arrays, and it is finally concluded that there is a relation between the pattern geometry and its absorptivity capacity; however, this requires further investigation. The simulated model and resultant prototypes are illustrated in Figure 11(b1) and (b2), respectively.

Another example is [134], where a flexible metamaterial absorber is manufactured via inkjet printing using silver nanoparticle inks in paper. One particularity is the design of the array, which is made of unit cell of a Jerusalem cross-resonator. The prototype is coated on a PET cylinder. The proposed design has not only the advantage of being eco-friendly and extremely light due to the zero chemical waste produced and the usage of paper substrate but also an excellent absorption rate thanks to the high conductivity of the ink. Likewise, ref. [135] introduces a modified split-ring resonator (MSRR) structure inkjet printed with silver metallic nanoparticle ink on a very low-cost photo paper. The objective of the article is to design an Xi-shaped left-handed metamaterial, where the unit cell design is much simpler: a 12 × 12 mm^2^ square made of combined SRR and CLS (capacitive loaded strip). The article proves being able to measure any loss in the double negative region, which is an interesting feature for X-band applications, especially for biomedical applications, such as for example in breast tumor detection. Figure 11 exposes the unit cell design (c1) and the bended prototype (c2).

**Figure 11 sensors-23-09103-f011:**
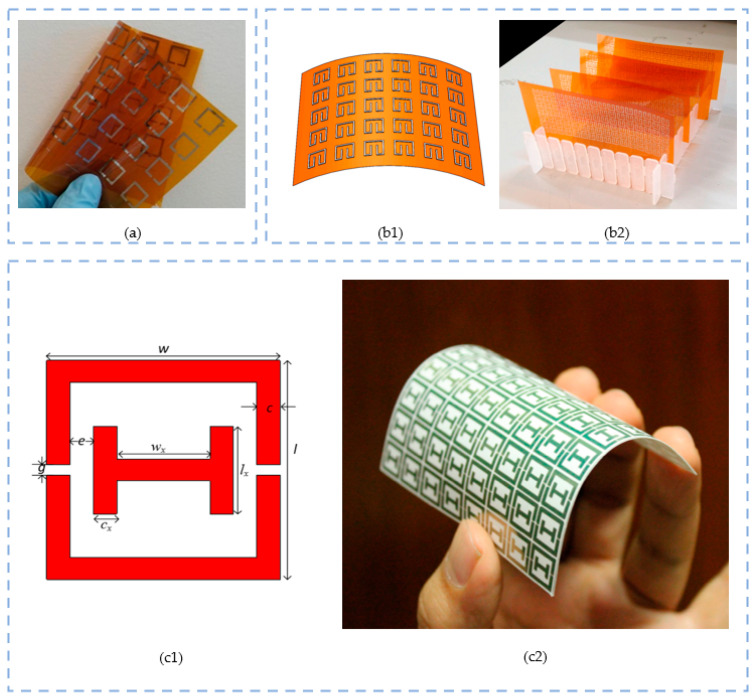
Some examples of printing technique for the design of resonators: (**a**) inkjet-printed array of SRR on a commercial polyamide film [132]; (**b1**,**b2**) inkjet-printed silver rectangle-based SRR resonator array on flexible metamaterial PI film [133]: layout (**b1**) and prototypes (**b2**); (**c1**,**c2**) inkjet-printed MSRR unit cell array on a very low-cost photo paper [135]: combined SRR and CLS unit cell (**c1**) and bended prototype (**c2**).

### 5.3. State-of-the-Art of Textile-Based Resonators for Wearable Applications

#### 5.3.1. Review of Different Resonator Topologies and Their Applications in Textile

Over the recent two decades, the interest in the usage of textile-based wearable devices for on-body applications has significantly increased, especially in the health monitoring and biomedical fields. As already mentioned in the article, microstrip antennas and resonators offer numerous advantages in terms of lightweight, small dimensions, geometrical possibilities, low manufacturing cost, durability, reliability and radiation innocuousness. Table 11 summarizes some articles found in the literature about resonators implemented onto textiles for different applications. Figure 12 below exposes some examples of resonators fabricated onto textiles.

#### 5.3.2. Metamaterials in Textiles

Metamaterials (MTMs) are materials whose artificial structures have been engineered to attain some properties which are infrequently observed in natural materials. The field of metamaterials provides alternative structures that grant noteworthy benefits in wireless applications, e.g., health monitoring, RFID tags, electromagnetic wave absorbers, filters and sensors, amongst others. In regard to the implementation of MTMs into resonators for microwave applications, many articles of interest are found in the literature, but only a few of them are truly oriented to its usage into textiles.

Table 12 describes some examples found in the literature of metamaterials implemented in textiles. The equivalent circuits are the same as the ones described in Section 2.2.1 (Figure 4).

It is a fact that MTM technology is revolutionizing the way we understand the wireless body networks and their application in clothing for many purposes with the biggest interested party being the health-monitoring sector.

#### 5.3.3. Evolution of the Embroidery Technique in the Design of Resonators

The first reference to the embroidery technique employed in resonators dates from 2009 [144], under the objective of designing a wearable textile barrier against microwave radiation, whereon a symmetrical array of uniformly oriented copper-wire thread C-shaped (split) rings is embroidered with electroconductive threads in twain pure flax (LI) and polyester (PES) woven textile samples. In the article, it is demonstrated that the obtained X-band attenuates significantly the microwaves within the frequency range of 7 to 10 GHz in spite of the non-uniformity in the pattern design.

Similarly, a flexible textile-integrated band-stop Frequency Selective Surface (FSS) is presented in [145] as a leveraged solution due to the flexibility, washability and scalability compared to conventional rigid substrate-based systems. In the literature, previous studies around the FSS principle on textiles are found [146,147]. In particular, ref. [148] presents an FSS-based textile-reflectarray (FSS-TRA) first prototype consisting of a combination of two layers—made of different fabrics—with a circular patch pattern design embroidered with conductive thread. This proposes a flexible, rollable and lightweight alternative for many satellite communication applications. In [145], the new FSS design consists of a unit cell being the top layer comprising a squared ring resonator pattern. It is manufactured in a multilayered loom substrate made of warp and weft threads using two different thermoplastics: polyethersulfones, PES, and polyethylene terephthalate, PET. A Shieldex^®^ conductive yarn is used to embroider the square rings in the top layer of the prototype. Finally, various set-ups are simulated and compared to an experimental 9 × 9-unit cell prototype, concluding in a stable performance in all scenarios, even under a condition of bending (radius of curvature), where the *S*_21_ level is over −10 dB out of the work bandwidth area (the model is meant to operate at a central frequency of 5 GHz with a bandwidth of 1.8 GHz).

The possibility of building wearable devices that can adapt to curve and wrinkled surfaces opens up a new line of investigation toward the body area network (BAN) applications. S.H.H. Mashhadi et al. [149] published the results of a study performed in multiple textile-based DRA prototypes set on a full-body dry phantom, which are compared to the simulations previously performed in CST MSW Software. In this article, a microstrip feed is embroidered with aluminum fiber conductive yarn on a conventional cotton cloth layered with buckram squared pieces all over a hard foam surface to enhance the stability of the assembly. The experimental results suggest a satisfactory performance given the reflection coefficient *S*_11_ of −10 dB in a range from 4.8 to 6.8 GHz approximately. From here, various configurations of the antenna are mounted in a mannequin, and the forward transmission wave |*S*_21_|^2^ (dB) power levels are analyzed to obtain the optimal position of the antennas in the body for on-body communication applications.

Multiple articles are found in the literature about the employment of a fully integrated embroidery technique involving resonators for the design of wearable antennas, filters and sensors. The most simplistic prototype is an antenna sensor based on an open ring resonator (ORR) excited through a microstrip line [12] for garments for wireless systems, which is manufactured using an embroidering machine, as seen in Figure 13a. The *S*_11_ parameter is tested in the sample under bending stress, and experimental results prove a relation between the bending radius and the resonance frequency, which is expressed in terms of sensitivity.

In [17], a similar study takes place. In this case, a split (squared)-ring resonator (SRR) is utilized in order to filter the signal propagation in UHF. Two models are presented in the article: first, an SRR, and after, a symmetrical double-SRR. In both cases, these are excited through the same MTM transmission line dimensions. For the prototyping, a limitation in the embroidery machine implies the combination of two different threads, which is taken into account when simulating it in CST Microwave Studio. A bending test is also performed in this study, this time analyzing the *S_21_* parameter. By the time the geometrical length is altered—actually reduced as a consequence of the bending—an evident frequency shift (144 MHz) in the upper direction occurs. The bending prototype using the single SRR and its effect on the resonance frequency/*S*_21_ at different bending radii are given in Figure 13(b1) and (b2), respectively.

Given the limitations in terms of precision of the embroidery machine, it is usually preferred to work with squared and rectangular shapes rather than curves. Nevertheless, embroidered circular split-ring resonators (SRRs) [138] and double-ring resonators (DRRs) [14] are also found in the literature. An example of a double-circular SRR is illustrated in Figure 13c. A recent (2019) innovative study [150] introduces the usage of an embroidered rectangular SRR wrapped around a low-density polyethylene (LDPE) bottle filled with a deionized (DI) water sample mixed up with four different solvents—ethanol, methanol, isopropanol and acetone—for its characterization. The measurement method consists of two monopole patch antennas that excite the resonator glued to the bottle filled with the DI + solvent sample. The *S*_21_ parameter is measured and compared with the CST Studio Suite simulations each time. Knowing the nominal resonance frequency of the empty bottle, the dielectric constant of the solvents can be determined by analyzing the frequency shift in each case. There is an agreement between the simulations and the experimental results, which determine a clear inverse relation between the relative permittivity value of the sample and the resonance frequency obtained.

To date, many other articles have been published proving the performance of embroidered resonators and antenna-based sensors in different textiles and subjected to several tests to evaluate the potential applications. A very interesting line of investigation is open to enhance the existing e-textile solutions that are foreseen to bring many advantages compared to the conventional techniques.

### 5.4. Challenges in Implementing Resonators for Flexible Electronics Design

Despite the already described advantages of using textile substrates in the design of wearable devices, there are some limitations and challenges that must be considered and well evaluated during the modeling phase. Washability and bending are addressed as the two principal characteristics of flexible electronics, but these are unfortunately difficult for many materials due to their nature. On top of these two features, there are other challenges to contemplate when designing wearable/flexible electronic devices.


**Power Transfer in Battery-Powered Devices**


In this article, the literature discussed so far refers to passive electronics applications only. However, one the principal dilemmas in the integration of battery-powered wearable devices is actually the dimensions of the battery.

In [151], a magnetic resonant coupling wireless power transfer (MR-WPT) system is proposed as an innovative solution. Although the transfer efficiency appears to be a limitation for many applications, it is well demonstrated that this system brings additional benefits in comparison to the conventional battery-based device method: for instance, the safety of magnetism for on-body applications. The design consists of a miniaturized symmetric four-resonator MR-WPT system prototype based on a textile substrate. Four different models of planar textile resonators (PTRs) are modeled, combining either copper tape or silver paste for the conductive coil pattern into two possible substrate textiles: polyester and cotton. Multiple thicknesses and transfer distances are also taken into consideration as variables to obtain the highest transfer efficiency (%) as well as to calculate the relation between the loss tangent, *tanδ*, of the textile substrates and the *S*_21_ (dB) parameter to further explain the resultant transfer efficiency values. In order to test the effectiveness of the models in the human body, two acrylic-based frames filled with muscle-mimicking liquid are used to represent the forearm and thigh of an adult man.

In conclusion, the highest transfer efficiency is obtained with the polymer substrate model, being an interesting candidate for wireless powered wearable devices essentially for medical or general clothing applications.


**Performance, Materials and Dimensions**


Industry demands are becoming more and more stringent when it comes to the performance and dimensions of flexible electronic devices. In order to reach a compromise between these two features, multilayered structures are suggested: vertically stacking (in the *Z*-axis) an arrangement of ring resonators multiplies the propagation field, meaning that the same resonator in the *XY* axis enhances its performance without rescaling its sizes. This is not only an advantage in terms of compactness, but also the transmission spectra are optimized. An example for the same is [152], where a two-layer stack of closed-ring resonator (CRR) arrays is numerically simulated using CST Studio Software. Multiple scenarios are presented under the objective of ascertaining whether it is possible to tune the resonance frequency by laterally displacing the stacked structure, and the results are promising: a resonance frequency shift occurs when relocating the stacked CRR arrays, where the frequency side depends on the direction of this displacement; and a dimension change—increasing the length and reducing the width—of the ring results in a heightened spectra shift.

Based on several articles found in the literature, it is confirmed that stacked multi-ring resonator (SMRR) structures have higher electromagnetic field coupling compared to conventional planar resonator structure; therefore, the Q-factor increases and the losses decrease. For this reason, these are perfect candidates for many applications such as the dielectric characterization of liquid and paste materials in the food, healthcare or agriculture industries, amongst others [153].

However, not many geometries have been achieved so far due to the limitations of arranging the coupling of the stacked resonators in the Z-dimension [154]. In addition to this, the materials and manufacturing techniques are also limited, since there must be a proper adherence between the substrate and the multiple layers of resonators, which has not been yet achieved in textiles. All of the above-mentioned points suppose additional challenges for implementing such innovative solutions to flexible electronics.


**Operability on Water**


As already mentioned in this article, external conditions such as temperature and humidity may alter the sensing device performance negatively. For this reason, it is crucial not only to acknowledge such conditions where the device is being used but also to determine the expected robustness of the sensor against water presence, especially if these are to be applicable in biological environments.

In [155], a 3D-printed CSRR-based sensor on a flexible Kapton substrate is proposed to overcome the performance issues when operating in contact with air and water droplets. It is reported that the prototype is suitable for its usage in sweat monitoring, humidity sensors or even for edible sensors applications. Not only water but also temperature and bending tests are performed. In all cases, optimal values of sensitivity and Q-factor are obtained. Henceforth, this study suggests a new line of investigation toward the development of waterproof and bendable sensing devices with high accuracy, which can be massively produced through 3D-printing technology.

## 6. Conclusions and Future Research Directions

The increasing demand of wearable devices for on-body applications and the more and more specific requirements in terms of performance, characteristics and applications has raised the interest of the scientific community and the industry regarding the usage of resonators as the core element for wireless communication and sensing applications—and more importantly for e-textiles development.

Although textiles are not electrically ideal materials, these offer good mechanical properties and other advantages such as flexibility, lightweight, washability, variety, versatility, being easy to work with, and many others. However, there are still multiple drawbacks to overcome, which are directly dependent on the materials’ nature and the dimensional limitations.

So far, the investigation has been mainly focused on the design of textile/flexible-based resonator sensors and their applications in the field of healthcare, communication and sports and leisure. Not much literature has been found about the operability of resonators embedded in textiles for microwave applications.

With all this being said, a list of key challenges and objectives is presented for future research in this area:Improving the performance of microstrip resonators for microwave applications by defining the design principles, selecting the optimal materials and conditions and outlining the manufacturing techniques.Introducing new materials (substrate and conductive) and investigating the novel techniques, especially embroidery and printing.Miniaturizing and optimizing the models and simplifying the fabrication process.Studying and proposing solutions to the main drawbacks (washability, bending).Expanding the possible scenarios and implementing the prototypes for realistic analysis, considering reliability tests, SAR measurement and operability on water as prerequisites for the design.

Textile wearable devices are already a reality, but there is a long path of improvement to provide a solution to the existing limitations nowadays. Further research is required to acknowledge the materials’ properties and introduce new methodologies that may serve for particular applications or simply to improve the sensing devices’ performance.

The reported research in the scientific literature toward resonators is mainly oriented to the materials’ characterization for academic or very specific scenarios, which represents a minor part of the possibilities that these elements offer in the sensing and filtering fields. Many parameters and variables can be combined to create multiple scenarios that would suggest promising opportunities for new models of textile-based devices in the near future.

## Figures and Tables

**Figure 1 sensors-23-09103-f001:**
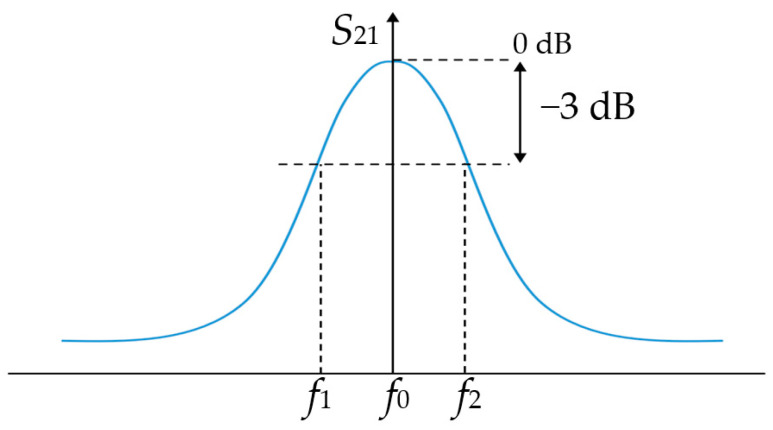
Graph of a bandpass filter Q-factor.

**Figure 2 sensors-23-09103-f002:**
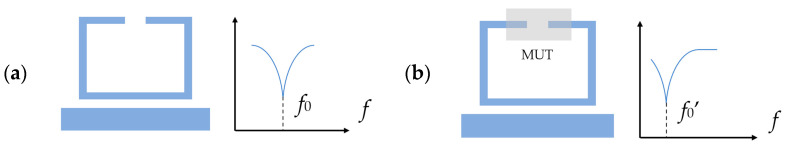
Frequency shift from (**a**) without MUT to (**b**) with MUT in the sensing area.

**Figure 3 sensors-23-09103-f003:**
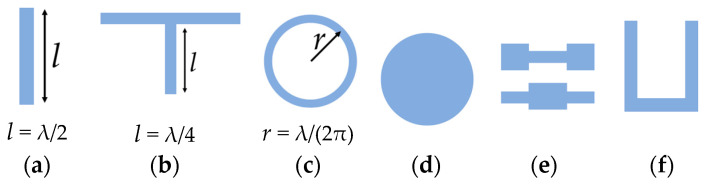
Some microstrip resonator structures: (**a**) half-wavelength (*λ/2*) resonator; (**b**) quarter-wavelength (*λ*/4) open-end stub resonator; (**c**) ring resonator; (**d**) circular patch resonator; (**e**) stepped impedance resonator; (**f**) hairpin resonator.

**Figure 4 sensors-23-09103-f004:**
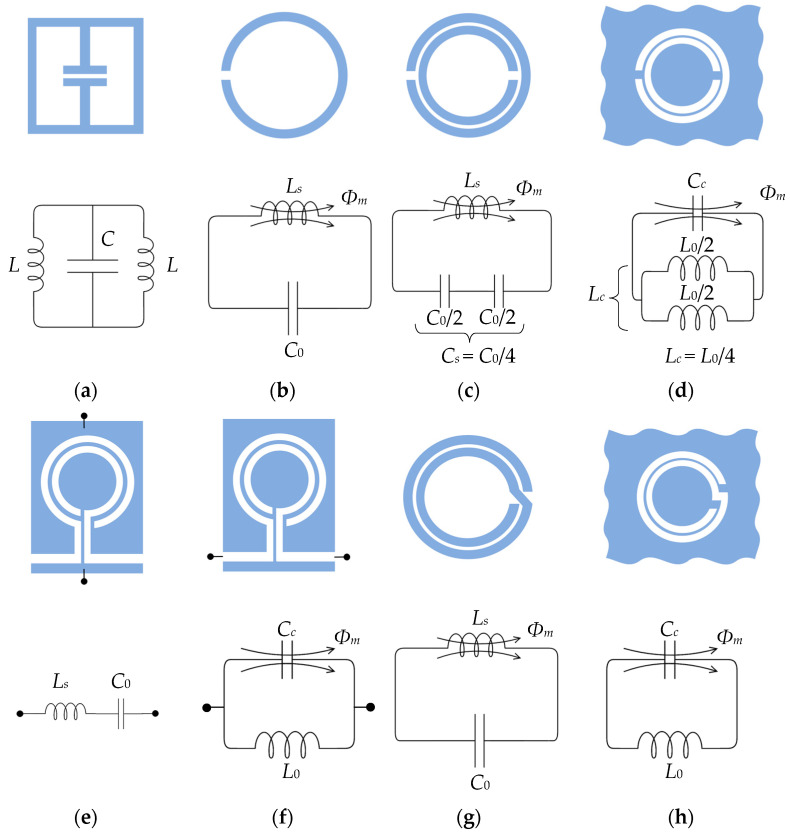
Topologies of the (**a**) ELC; (**b**) single SRR; (**c**) double SRR; (**d**) double CSRR; (**e**) OSRR; (**f**) OCSRR; (**g**) SR; (**h**) CSR; and their equivalent circuit models. The ohmic losses would be considered by adding a series resistance in the model. Models reproduced from references [3,21].

**Figure 5 sensors-23-09103-f005:**
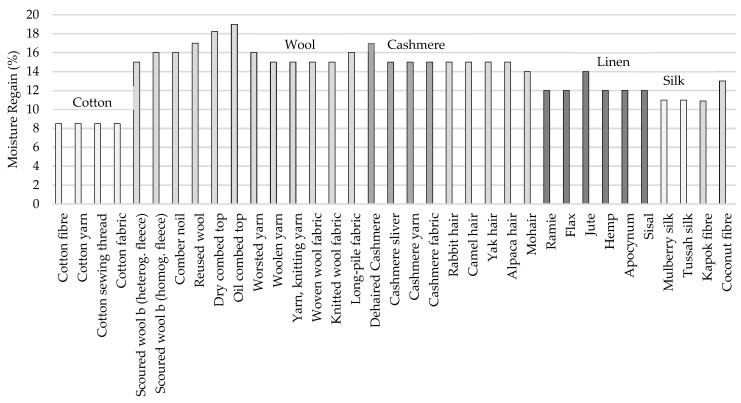
Moisture regains (%) of natural fibers [31,32,33].

**Figure 6 sensors-23-09103-f006:**
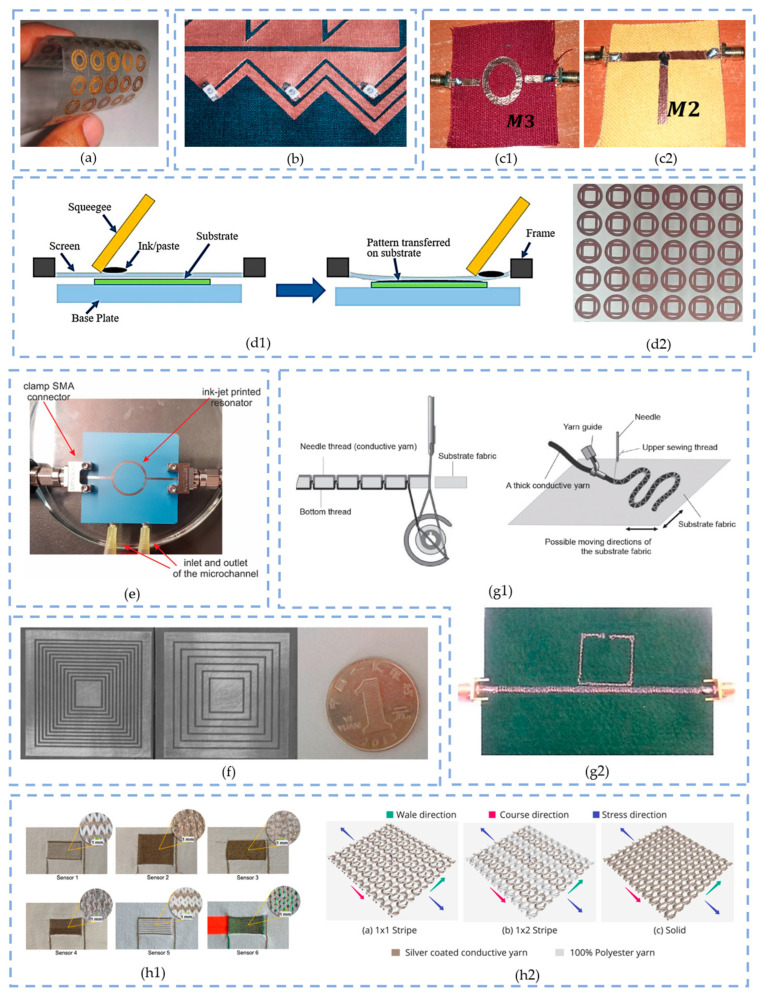
Some examples of wearable electronics manufacturing techniques: (**a**) Wet Etching: flexible split-ring resonator metamaterial structure fabricated on a polypropylene film by chemical etching [94]; (**b**) Welding and Soldering: components surface-mounted soldered to conductive fabric. Reproduced with permission. Copyright Liza Stark (http://thesoftcircuiteer.net/); (**c1**,**c2**) Adhesive Conductive Foil: implementation of (**c1**) a ring resonator and (**c2**) a stub resonator on textiles using adhesive copper tape [29]; (**d1**,**d2**) Screen Printing: (**d1**) an illustration of a conventional screen-printing process; (**d2**) prototype of 60 GHz flexible meta surface made of a unit cell of circular metallic rings embedded with a square-shaped ring and printed on Melinex [95]; (**e**) Inkjet Printing: split-ring resonator fabricated using an ink-jet printer using ink with silver nanoparticles [77]; (**f**) 3D Printing: chipless RFID tag made of rectangular slot ring tags, different IDs (left to right): ‘1111111111110’ and ‘1010101010100’ [96]; (**g1**,**g2**) Embroidery: (**g1**) embroidery methods (from left to right): standard embroidery and TFP [92]; (**g2**) embroidered transmission line loaded with a split-ring resonator on felt substrate with a satin pattern (60% density) [97]; (**h1**,**h2**) Weaving and Knitting: (**h1**) different knitted plain structure-based sensors and (**h2**) illustrated simulation of plain knit structure composed of conductive yarn and non-conductive yarn made from 100% polyester [93].

**Figure 7 sensors-23-09103-f007:**
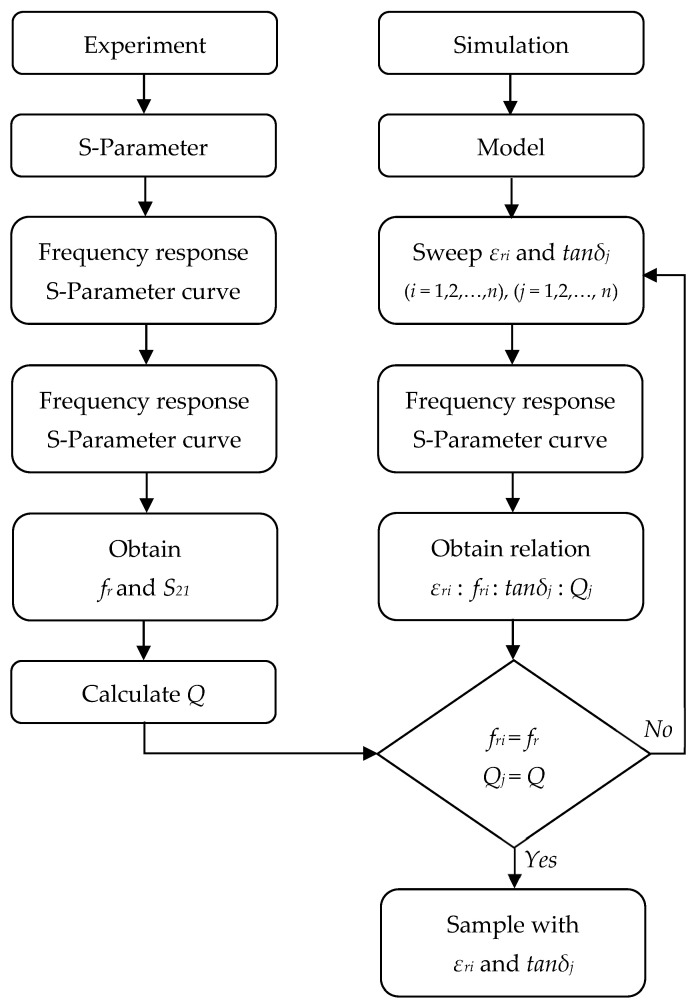
Flow chart for the deviation of material’s dielectric characterization.

**Figure 8 sensors-23-09103-f008:**
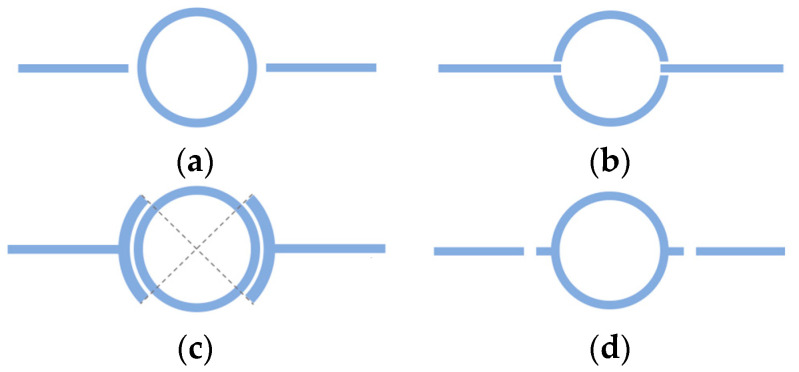
Conventional resonator coupling methods: (**a**) loose coupling; (**b**) enhanced coupling; (**c**) line-to-ring coupling; (**d**) matched-stubs coupling.

**Figure 9 sensors-23-09103-f009:**
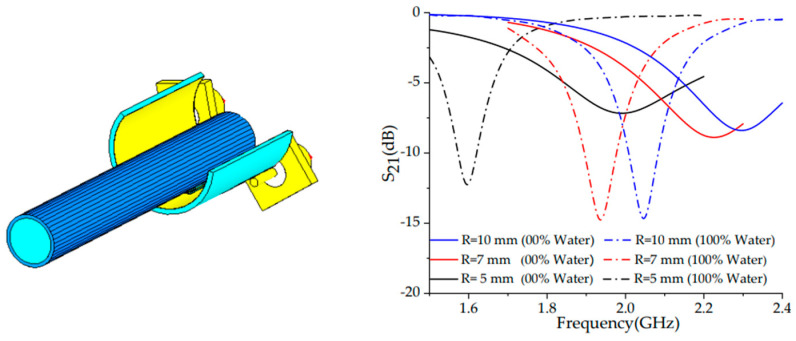
Microwave liquid sensor based on a metamaterial CSRR: (**left**) design of the sensor using CST software and (**right**) *S*_21_ response of water–ethanol mixtures for different concentrations and bending radii [108].

**Figure 10 sensors-23-09103-f010:**
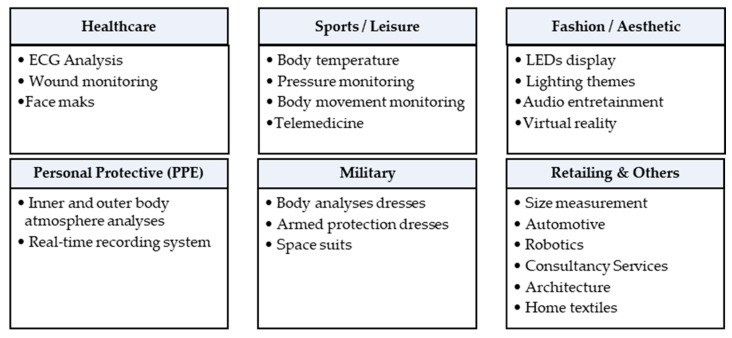
Summary of applications of e-textiles wearable technology.

**Figure 12 sensors-23-09103-f012:**
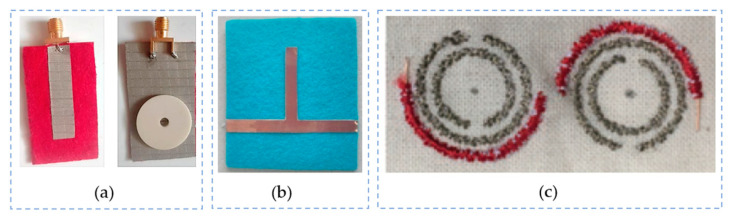
Some examples of fabricated resonator-based sensors for textile applications: (**a**) dielectric resonator antenna button textile antenna for off-body applications [136]; (**b**) stub resonator in woolen felt substrate [137]; (**c**) embroidered SRR for antenna design [138].

**Figure 13 sensors-23-09103-f013:**
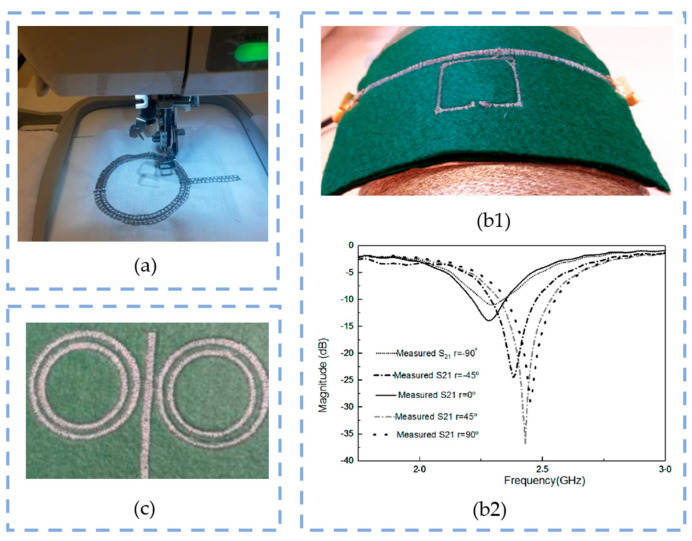
Some examples of embroidery technique used in the design of resonator-based sensors and antennas: (**a**) embroidery of an ORR-based antenna sensor [12]; (**b1**,**b2**) fully embroidered transmission line loaded in a squared SRR on a felt substrate [17]: (**b1**) bended prototype and (**b2**) study of the bending effect on the frequency and *S*_21_ parameter; (**c**) embroidered double-circular SRR prototype [14].

**Table 1 sensors-23-09103-t001:** Configurations of lossy transmission lines and their *unloaded Q* at the resonance frequency [19].

Transmission Line Configuration	Length *l* at Resonance ^1^	Unloaded *Q* (*Q*_0_) ^1^	Input Impedance *Z_in_* at Resonance Frequency ^2^	Circuit Form
Short-Circuited λ/2 Line	l=πβ	Q0=π2αl	$Z_{in}=R+2jL\Delta \omega$	Series circuit
Short-Circuited λ/4 Line	l=π2β	Q0=π4αl	Zin=1(1/R)+2jLΔωC	Parallel circuit
Open-Circuited λ/2 Line	l=πβ	Q0=π2αl	Zin=Z0αl+j(Δωπ/ω0)	Parallel circuit

^1^ Here, *l*, *α* and *β* represent the length (cm), the attenuation constant (Np/m) and the propagation constant (rad/m), in this order. ^2^ *R* and *C* represent the resistance (Ω) and the capacitance (F) of the equivalent circuit, in this order.

**Table 2 sensors-23-09103-t002:** Intrinsically soft materials for flexible electronics [24].

Layer	Applications	Materials
Substrate Materials	Polymers	PVA, PET, PI, PU, PE
	Flexible and stretchable	PEN, PDMS
	Eco-friendly and biodegradable	Paper, chitin, silk, gelatin
Electrode Materials	Soft metals	Galinstan alloy, metal nanoparticles and nanowires
	Conductive nanomaterials	Carbon materials: graphene, carbon nanotube
	Conductive polymers	PEDOT: PSS, PANI
	Hybris materials	Carbon/metal, polymer/metal
Functional Materials	Cluster materials	Molecules, polymers, biomass
	Low-dimensional and nanomaterials	Nanoparticles and Quantum dots Nanowires, nanotubes and nanoribbons Graphene, transmission metal dichalcogenides and mXenes
	Organic–inorganic hybrid materials	Nanomaterials/polymers

**Table 3 sensors-23-09103-t003:** Properties of substrates: metal foil, plastic film and thin glass [24,25,26].

Property	Metal Foil	Plastic Film	Thin Glass
Thickness, *t*	Stainless steel 430 100 μm	PEN, PI 100 μm	Glass 1737 100 μm
Flexibility	At t < 125 μm	High	At t < 100 μm
Young’s modulus, *E*	Stainless steel 430 200 gPa	PEN, PI 5 gPa	Glass 1737 70 gPa
Optical transmission visible	Opaque	PEN and PET semitransparent (optical transmittance >85%)	Semitransparent (optical transmittance >90%)
Resistance to process chemicals	High (Stainless steel)	Poor for PC, PES, PAR, PCO	Resistance to most of them
Dimensional stability	High	Lower than glass	High
Temperature tability	High	Lower than glass	High
Temperature tolerance	≤1000 °C	≈200 °C	≤600 °C
Thermal conductivity	Stainless steel 430 16 W/m°C	PEN, PI 0.1–0.2 W/m°C	Glass 1737 1 W/m°C
Electrical conductivity	Conductive Insulation requires insulating layer coating	Insulator (Except ICPs)	Insulator
CTE ^1^	Low (<20 ppm/°C) (Stainless steel 430, 304)	Low (<20 ppm/°C) (PET, PEN, PI)	Low (≈4 × 10^−6^ ppm/°C)
Surface roughness	≈1 μm	≈10 μm	≤1 μm
Durability	High	Lower than metal	Lower than metal
Permeability against oxygen and water	Impermeable	Easily permeated by oxygen and water	Impermeable
Roll-to-toll processable	Yes	Likely	Not likely

^1^ Coefficient of thermal expansion.

**Table 4 sensors-23-09103-t004:** Comparison of different materials characterized using ring resonator and patch antenna methods.

Ref.	Fabric	*ε_r_*	tan*δ*	Frequency (GHz)
[41,42]	Felt	1.22–1.38	0.0160–0.0230	2.45
[41]	Cordura^®^	1.90	0.0098	2.45
[41]	Cotton ^1^	1.60 ^1^	0.0400 ^1^	2.45
[41]	100% Polyester	1.90	0.0045	2.45
[43]	Viscose	1.64	0.0160	2.45
[41]	Lycra	1.68	0.0080	2.45
[41]	Quartzel^®^ Fabric	1.95	0.0004	2.45
[41]	Cordura/Lycra^®^	1.50	0.0093	2.45
[41]	Tween	1.69	0.0084	2.45
[41]	Panama	2.12	0.0500	2.45
[41,43]	Jeans	1.62–1.70	0.0180–0.0250	2.45
[41]	Moleskin	1.45	0.0500	2.45
[43]	Fleece	1.20	0.0040	2.40
[43]	Felt	1.36	0.0160	2.40
[43]	Silk	1.20	0.0250	2.40
[43]	Leather	2.95	0.1600	2.40
[43]	Neoprene rubber	5.20	0.0250	2.40
[43]	Cotton	1.54	0.058	2.40
[43]	Polystyrene foam	1.02	0.00009	2.40
[43]	Velcro	1.34	0.0060	2.40 & 5.00
[43]	Denim	1.80	0.0700	2.40 & 5.00
[43]	Tween	1.69	0.0084	2.60–3.95
[43]	Silk	1.75	0.0120	2.60–3.95
[42]	Polyester plain weave	3.20–3.66	0.018–0.0320	1.00
[42]	Flax	4.22–6.20	0.0370–0.058	1.00
[42]	Jute	2.99–4.95	0.0310–0.0470	1.00
[42]	Hemp	4.08–4.77	0.0280–0.0520	1.00
[42]	Wool	4.11–5.70	0.0280–0.0520	1.00
[44]	100% Nylon 6.6	2.82–2.78	0.0268–0.0283	9.80
[44]	Denim	1.70	0.085	2.22–2.59

^1^ More detailed table in [42].

**Table 5 sensors-23-09103-t005:** Fiber categories, properties and their main applications [47,48,49].

Fiber Category	Material	Properties
Metal fibers	Stainless steel Silver Copper Aluminum Nickel Many alloys	High conductivity (≈105 S/cm) High mechanical and thermal strength Heat and corrosion resistant Anti-static properties Electromagnetic shielding
Carbon fibers	Polyacrylonitrile (PAN) Pitch Viscose Carbon black Graphene	High conductivity (≈104 S/cm) High mechanical strength Large specific surface area Low mass density Excellent mechanical properties Abrasion and wear resistance Anti-static properties are possible also
Conducting polymers	Polyaniline (PANI) Polypyrrole (pPy) Polystyrene sulfonate (PEDOT: PSS) Polythiophene (PT)	Excellent flexibility Low conductivity (the non-ionic polymers only) Poor mechanical properties Good electronic properties: high areal capacitance and areal energy density

**Table 6 sensors-23-09103-t006:** Yarn categories and their main applications [52].

Yarn Category	Applications
Metallic	Fabric-based circuits Core of composite yarns Heating fabrics Antimicrobial textiles Antipiercing gloves Anti0static
Conductive composite (MCEYs)	Antistatic and electromagnetic shielding Antibacterial Strain sensor
ICP yarns	Electromagnetic shielding Simple circuit Supercapacitor
Plated, laminated and coated	Supercapacitor Strain sensor
New conductive and nanomaterial	Microelectrode Antibacterial Antimicrobial textiles

**Table 7 sensors-23-09103-t007:** Summary of the main advantages and challenges of the most common wearable electronics integration techniques.

Technique	Advantages	Limitations
Wet Etching	Controllable etching rate Complex designs High resolution	Chemical contamination Depends on orientation Undercutting Not for small batch production Influenced by temperature and concentration of etchant High costs
Soldering and Welding	Low contact resistance Easy manufacture Efficient mass production Low cost Compatible with standard electronics processes Ultrasonic welding: not contaminant, biocompatible	Mechanically fragile Requires reinforcing when bending or stretching (not flexible at all) Requires high temperature Limited materials
Adhesive Conductive Foil	Low cost In-house made Flexible Good for prototyping Low curing temperatures Environmentally friendly	Disengages easily Low accuracy Not suitable for complex designs Fragile, easy to break Poor resolution High contact resistance
Inkjet Printing	No printed form is needed Ideal for prototyping High-quality prints Small losses	Limited colors Elevated cost of inks Poor print quality Slow print
Screen Printing	Price competing Good ink processability Efficient mass production Applicable to a final product Very flexible and stretchable Ideal for simple design	Printed form is required Long set up time Limited layer thickness Not efficient for prototyping or small batch production Huge waste of water Not possible in-house
3D Printing	In-house made Fully digital Fast fabrication Flexibility in materials Complex shapes and structures Adaptable density Lightweight	Difficult scalability High cost of AM manufactured material per weight Inconsistent quality Lossy materials Thermal instability Bending difficulties Poor resolution
Embroidery	2D and 3D structures Freedom of fibers orientation Good for prototyping Reduced waste Stable textile structures Very flexible and stretchable Precise and digitized (CAD) Variety of materials Scalability, mass production	Low precision in complex designs Difficult to automate Requires sealing to avert electrical shorting Slow process Performance impacted when bending or washing
Weaving and Knitting	Highly automated Stable textile structures Robust textile structures	Requires sealing to avert electrical shorting Fragile, easy to break Limited choice of yarns Performance impacted when bending or washing

**Table 8 sensors-23-09103-t008:** Comparison of resonance methods [42,98,99,100,101,102,103,104].

Measurement Method	Brief Description	Materials States	Materials Assumptions	Dielectric Properties	Measure Parameters	Frequency Range	Propagation Mode	MUT Loss Factor	Advantages	Drawbacks
Open-ended coaxial line (coaxial probe)	Flat probe on the surface of the solid or immersed in the liquid to sense reflected signal	Liquids Semi-solid	Semi-infinite thickness Non-magnetic Isotropic Homogeneous Flat surface No air gaps	*ε_r_*	*S* _11_	Broadband (RF, MW) (200 MHz–20 GHz, even 100 GHz)	TEM ^1^ or TE ^2^	Lossy	Accurate Easy manufacturing No sample preparation Non-destructive	Errors due to air gaps Air bubbles Repetitive calibration Large samples require flat surfaces Low loss resolution
Transmission/reflection line (waveguide and coaxial)	Sample placed inside the transmission line filling its cross-section Generates impedance change	Liquids Solids	Flat face Perpendicular to long axis Homogeneous No air gaps	*ε_r_*, *µ_r_*	*S*_11_, *S*_22_ or *S*_12_, *S*_21_	Broadband (<100 GHz)	TE for waveguides TEM for coaxial	Lossy Low loss	More accurate than probe method High sensitivity	Less accurate than resonators Destructive Complex preparation Sample length limitation
Free space	Sample placed between Rx/Tx antennas Attenuation and phase shift are measured	Solids	Flat face Parallel-faced Homogeneous or inhomogeneous	*ε_r_*, *µ_r_*	*S*_11_, *S*_21_	Banded (MW) (300 MHz–300 GHz)	TEM	High loss Low loss	Non-contacting Non-destructive High temperature	Requires special calibration Large and flat MUT Diffraction at the edges
Parallel plate (electrode)	Sample is placed between two electrodes, forming a capacitor	Thin surface sheets	Thin, flat sheet sample	*ε_r_*	*S*_11_, *S*_21_	Low frequencies (<100 MHz)	TEM or TE	High loss	Very accurate Cost-effective	Limited frequency band Delicate and thin sample (<10 mm)
Resonant cavity (cavity resonator)	Loaded sample introduced in a cavity altering the resonance frequency	Solids	Small samples High impedance	*ε_r_*, *µ_r_*	Frequency Q-factor	Single (1 GHz–30 GHz)	TE or TM ^3^	Low loss	Easy preparation Very accurate High temperature	Destructive Small samples Single or resonance frequency only Complex measurement
Planar transmission line loaded resonators	Sample placed on top of the resonator interfering with the substrate or air-boundary	Liquids Solids	Substrate (solid) Superstrate (solid and liquid)	*ε_r_*, *µ_r_*	*S*_11_, *S*_21_	Single (MW) (<100 GHz)	TEM or Quasi-TEM	Low loss	Very accurate Cost-effective High temperature Easy manufacturing	Errors due to air gaps Low Q-factor

^1^ Transverse magnetic mode, ^2^ Transverse electric mode, ^3^ Transverse electromagnetic mode.

**Table 9 sensors-23-09103-t009:** Dielectric measurement methods according to the material’s properties and the sample preparation [105].

Measurement Method	Material Category	Accuracy	Fixture Type	Specimen Preparation	Material Loss
Waveguide Resonator	Bulk	High	Discrete Frequencies	Complex	0.01 < *tanδ* < 0.2
Transmission Line Resonator	Bulk	High	Discrete Frequencies	Complex	0.01 < *tanδ* < 0.2
Split-Post Resonator	Bulk	High	Discrete Frequencies	Simple	0.0005 < *tanδ* < 0.02
Split-Cylinder Resonator	Bulk	High	Discrete Frequencies	Simple	0.0005 < *tanδ* < 0.02
Re-entrant Cavity	Bulk	High	Discrete Frequencies	Complex	0.0005 < *tanδ* < 0.02
Dielectric Resonator	Bulk	High	Discrete Frequencies	Complex	0.0005 < *tanδ* < 0.02
Thin-Film Resonator	Thin Film	High	Discrete Frequencies	Complex	0.01 < *tanδ* < 0.2
Evanescent Probe	Thin Film	High	Discrete Frequencies	Simple	0.01 < *tanδ* < 0.2
Multiple Transmission Lines	Bulk	Low	Broadband	Complex	0.01 < *tanδ* < 0.2
Filled Waveguide	Bulk	Low	Broadband	Complex	0.01 < *tanδ* < 0.2
Filled Transmission Line	Bulk	Low	Broadband	Very Complex	0.01 < *tanδ* < 0.2
Parallel Plate Capacitor	Bulk	Low	Broadband	Simple	0.01 < *tanδ* < 0.2
Open Coaxial Probe	Bulk	Low	Broadband	Very Simple	0.01 < *tanδ* < 0.2
Thin-Film Transmission Lines	Thin Film	Low	Broadband	Complex	0.01 < *tanδ* < 0.2
Thin-Film Capacitors	Thin Film	Low	Broadband	Complex	0.01 < *tanδ* < 0.2

**Table 10 sensors-23-09103-t010:** Impact of washing on electronic textile-based devices.

Ref.	Resonator/Antenna Design	Conductive Material	Substrate	Washing Cycles	Radiation Coefficient, *S*_11_	Radiation Efficiency, η	Frequency Shift (GHz)
[62]	Rectangular microstrip ring resonator antenna	Copper sheet metal foil with synthetic (resin) adhesive	Wool	1	Significant increase (≈11 dB) at *f_r_* ^1^	Minor variation	<0.5
			Corduroy	1	Minor decrease (≈4 dB) at *f_r_*	Minor variation	<0.5
			Jeans	1	Minor increase (≈1 dB) at *f_r_*	Minor variation	1.5 to 2.5
[110]	Microstrip inset-fed patch antenna	Silver-based ink 1 Electrodag	Cotton/Polyester	5	Minor increase (≈1 dB) at *f_r_*	Increased	<0.2
			Polyester	5	Minor decrease (≈1 dB) at *f_r_*	Increased	<0.2
		Silver-based ink 2 DuPont	Cotton/Polyester	5	Minor decrease (≈1 dB) at *f_r_*	Increased	<0.2
			Polyester	5	Minor increase (≈1 dB) at *f_r_*	Increased	<0.2
[111]	Dual-band coplanar patch antenna	Woven conductive fabric ‘Zelt’	Felt	1	Unknown Gain reduced by ≈3 dBi	No variation	5% downwards
[112]	UHF RFID Tag	Screen printing polymer thick film (PTF) ink glued with multiple coatings	100% cotton	1	Unknown Gain decreased by 1.5 dBi	Unknown	Negligible (10^−3^)
			35% cotton/65% polyester	1	Unknown Gain decreased by ≈10 dBi	Unknown	Negligible (10^−3^)

^1^ Resonance frequency.

**Table 11 sensors-23-09103-t011:** Summary of common topologies of resonators for wearable applications in textiles.

Topology	Ref.	Description	Applications	Advantages	Challenges
Dielectric Resonator Antenna (DRA)	[136]	Ring-shaped DRA on textile coupled to microstrip line	Off-body: reduce the SAR ^1^ for WLAN applications	Large bandwidth Small dimensions Many geometries and feeding techniques are possible Low profile and high gain	Miniaturization limitations Inaccurate experimental results due to unprecise design
Stub Resonator	[137]	T-resonator made of metallic tape on a woolen felt substrate	Dielectric characterization of materials at microwave and high frequencies	Simple, fast and low-cost prototyping Good choice for academical research only	Inaccurate results compared to simulation
	[139]	3rd L-shaped open stub *λ*/4 resonator connected to a microstrip transmission line and to a UWB ^2^ antenna. Silver screen printing onto a coated nylon fabric	Textile-based chipless RFID tag used to compete with the conventional clothing barcode	Good precision and quality in the design Simple and fast manufacturing technique Potentially supports multiple washing cycles	Dimensional limitations compared to the miniaturized PCBs used nowadays Thin material may impact the performance
Ring Resonator	[140]	SRR wearable reader antenna made of flexible adhesive electro-textile EPDM ^3^ material-based substrate	Body-worn antenna: wearable UHF RFID reader integrated with work gloves	Simple design (single-layered structure) Easy integration Accurate results compared to simulations	Not as stable as the slotted patch antenna also studied in the article

^1^ Specific absorption rate, used to quantify the thermal effect (energy absorption levels) in the human tissue. ^2^ Ultra-wide band. ^3^ Ethylene propylene diene monomer.

**Table 12 sensors-23-09103-t012:** Summary of five articles of metamaterials in textiles found in the literature.

Ref.	MTM Design	Main Characteristics and Conclusions
[141]	Decagonal C-shaped CSRR textile-based MTM	Pioneer and unique flexible and negative index structure Felt substrate with MTM unit cell of ShieldIt Super^TM^ conductive textile MTM model can be employed for wireless applications
[142]	MTM textile network (complex structure)	Power splitters, antennas and a split-ring resonator Made of laser-cut conductive textile (Cu/Ni polyester) and pasted into fabric with adhesive The MTM textile has the capability to set up an efficient wireless connection, e.g., Bluetooth, to transmit health data wirelessly to a smartphone in a short distance (<10 cm), to transfer wireless power and to react to touch sensing
[143]	Compact chipless RFID MTM multi-CSRR structure	Multi-resonator structure made of 4 rectangular slotted CSRRs and a planar transmission line on a fleece substrate The main objective is to miniaturize the size Analysis of the effects of the orientation, number of elements and substrate types The chipless RFID tag is optimized by using a low loss material and adding ring resonators
[17]	Embroidered MTM transmission line for signal propagation control	Fully embroidered transmission line loaded with multiple SRRs on a felt substrate It is possible to control and filter the UHF signals: stop-band levels over −30 dB are achieved Embroidery turns out to be a proper technique to employ given the good agreement between experiment results and simulations Bending is tested and frequency shift is measured
[97]	Textile MTM performance for wireless body area network systems	Fully embroidered transmission line loaded with conductive yarn SRRs Felt and cotton fabrics are used as substrates and compared The objective to control the signal propagation is well achieved: stop-band filter with a high level of signal rejection

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
