# Peer review of "Microwave Resonators for Wearable Sensors Design: A Systematic Review"

_sensors, 2023, doi:10.3390/s23229103_

Round 1

Reviewer 1 Report

Comments and Suggestions for Authors

In this paper the authors present a review about microwave resonators for wearable sensors. The developed analysis is quite complete since they consider different geometries, materials, fabrication processes and so on. 

Some improvements are needed before publication: 

-In the introduction section authors declare that in section 5 an analysis about antennas and filters has been reported. This part needs to be improved in the relative section, especially regarding antennas. A paragraph about SAR should be added in this context.

- Since the topic regards wearable sensor, it should be useful a reference regarding the ISM band which is largely used for this purpose. 

-Authors should add a section regarding biocompatible solutions in this field.

-More references of the last year need to be added. Here some on them: 

Nazli Kazemi, Mohammad Abdolrazzaghi, Peter E. Light, Petr Musilek, "In–human testing of a non-invasive continuous low–energy microwave glucose sensor with advanced machine learning capabilities", Biosensors and Bioelectronics, Volume 241, 2023.

E. Mansour, M. I. Ahmed, A. Allam, R. K. Pokharel and A. B. Abdel-Rahman, "A Microwave Sensor Based on Double Complementary Split-Ring Resonator Using Hexagonal Configuration for Sensing Diabetics Glucose Levels," 2023 17th European Conference on Antennas and Propagation (EuCAP), Florence, Italy, 2023, pp. 1-5, doi: 10.23919/EuCAP57121.2023.10133248.

I. Marasco, G. Niro, F. Rizzi, A. D’Orazio, M. Grande and M. De Vittorio, "Design and Fabrication of a Minimally Invasive Dielectric Sensor for Biological Environments," in IEEE Access, vol. 11, pp. 103452-103460, 2023, doi: 10.1109/ACCESS.2023.3317697.

Y. Zhang, M. Shafiei, J. Z. Wen, Z. Abbasi and C. L. Ren, "Simultaneous Detection of Pressure and Bending Using a Microwave Sensor With Tag and Reader Structure," in IEEE Transactions on Instrumentation and Measurement, vol. 72, pp. 1-11, 2023, Art no. 9511311, doi: 10.1109/TIM.2023.3288253.

-References should be written by following the journal style. Please revise.

Comments on the Quality of English Language

 Moderate editing of English language is required.

Author Response

In this paper the authors present a review about microwave resonators for wearable sensors. The developed analysis is quite complete since they consider different geometries, materials, fabrication processes and so on. 

Some improvements are needed before publication: 

  1. In the introduction section authors declare that in section 5 an analysis about antennas and filters has been reported. This part needs to be improved in the relative section, especially regarding antennas. A paragraph about SAR should be added in this context.

A paragraph about SAR is added at Section 5.1.1. under sub-section ‘SAR and Bending Reliability Tests of Flexible Resonator-based Sensors’, page 30 of 48. The reference to antennas and filters is removed from the introduction since it is misleading. The objective of the paper is to study resonators.

  1. Since the topic regards wearable sensor, it should be useful a reference regarding the ISM band which is largely used for this purpose. 

A brief mention to ISM band is added in Section 5.1, page 28 of 48, and referenced to:

  1. Kaur and P. Chawla, ‘Design and performance analysis of wearable antenna for ISM band applications’, Int. J. Electron., vol. 110, no. 6, pp. 986–1005, Jun. 2023, doi: 10.1080/00207217.2022.2068199.
  2. Authors should add a section regarding biocompatible solutions in this field.

We appreciate the reviewer’s remark. However, the first focus of this paper is on existing wearable resonators, and unfortunately the exploration field in biocompatible solutions is relatively new: the literature found is limited to few articles. For this reason, we consider keeping on investigating and enhancing the State-of-the-Art Review in the future including as much research as possible.

Literature found with Advanced Search through IEEE with key words ("All Metadata":wearable) AND ("All Metadata":biocompatible) AND ("All Metadata":resonator):

  • Rithika, T. J. Sweety, T. R. Arun, S. K and J. Jose, "CPW Fed SRR Inspired Decagon-Trident Slotted Shape Antenna for On-Body Communication," 2022 IEEE Microwaves, Antennas, and Propagation Conference (MAPCON), Bangalore, India, 2022, pp. 707-710, doi: 10.1109/MAPCON56011.2022.10046977.
  • Niro, I. Marasco, F. Rizzi, A. D’Orazio, M. De Vittorio and M. Grande, "Design of a surface acoustic wave resonator for sensing platforms," 2020 IEEE International Symposium on Medical Measurements and Applications (MeMeA), Bari, Italy, 2020, pp. 1-6, doi: 10.1109/MeMeA49120.2020.9137116.
  • Clausen, T. M. Seeberg, C. Gheorghe and F. Prieur, "Investigations of TiO2 as a protective coating on diaphragm-based in vivo sensors," Proceedings of the 6th International Workshop on Wearable, Micro, and Nano Technologies for Personalized Health, Oslo, Norway, 2009, pp. 21-24, doi: 10.1109/PHEALTH.2009.5754821.
  • Sajith, J. Gandhimohan and T. Shanmuganantham, "SRR loaded CB-CPW fed hexagonal patch antenna for EEG monitoring applications," 2017 IEEE International Conference on Circuits and Systems (ICCS), Thiruvananthapuram, India, 2017, pp. 66-70, doi: 10.1109/ICCS1.2017.8325964.
  • Sajith, J. Gandhimohan and T. Shanmuganantham, "Design of SRR loaded CB-CPW fed diamond shaped patch on-body antenna for ECG monitoring applications," 2017 IEEE International Conference on Circuits and Systems (ICCS), Thiruvananthapuram, India, 2017, pp. 118-123, doi: 10.1109/ICCS1.2017.8325975.
  1. More references of the last year need to be added. Here some on them:

Nazli Kazemi, Mohammad Abdolrazzaghi, Peter E. Light, Petr Musilek, "In–human testing of a non-invasive continuous low–energy microwave glucose sensor with advanced machine learning capabilities", Biosensors and Bioelectronics, Volume 241, 2023.

Reference [116] is included in the article and referenced in Section 5.1 under ‘Healthcare’ sub-section, page 27 of 48.

  1. Mansour, M. I. Ahmed, A. Allam, R. K. Pokharel and A. B. Abdel-Rahman, "A Microwave Sensor Based on Double Complementary Split-Ring Resonator Using Hexagonal Configuration for Sensing Diabetics Glucose Levels," 2023 17th European Conference on Antennas and Propagation (EuCAP), Florence, Italy, 2023, pp. 1-5, doi: 10.23919/EuCAP57121.2023.10133248.

Reference [117] is included in the article and referenced in Section 5.1 under ‘Healthcare’ sub-section, page 27 of 48.

  1. Marasco, G. Niro, F. Rizzi, A. D’Orazio, M. Grande and M. De Vittorio, "Design and Fabrication of a Minimally Invasive Dielectric Sensor for Biological Environments," in IEEE Access, vol. 11, pp. 103452-103460, 2023, doi: 10.1109/ACCESS.2023.3317697.

Reference [154] is included in the article and referenced in Section 5.4 under ‘Operability on Water’ sub-section, page 37 of 48.

  1. Zhang, M. Shafiei, J. Z. Wen, Z. Abbasi and C. L. Ren, "Simultaneous Detection of Pressure and Bending Using a Microwave Sensor With Tag and Reader Structure," in IEEE Transactions on Instrumentation and Measurement, vol. 72, pp. 1-11, 2023, Art no. 9511311, doi: 10.1109/TIM.2023.3288253.

Reference [122] is included in the article and referenced in Section 5.1.1 under ‘Bending Reliability Tests’ sub-section, page 29 of 48.

  1. -References should be written by following the journal style. Please revise.

References have been revised and corrected.

Reviewer 2 Report

Comments and Suggestions for Authors

This manuscript presents a review on multiple approaches for wearable resonators used as antenna-based systems, sensors and filters. This review includes advantages and drawbacks of the flexible resonators, materials, fabrication processes, and applications. This manuscript can improve based on the following comments:

1.- This manuscript can add a section about modeling or numerical simulations (e.g., finite element models) on the performance of flexible  resonators used as antenna-based systems, sensors and filters.

2.- The authors can improve their manuscript by considering a section or information on the reliability tests of flexible  resonators.

3.- The authors can add more applications or potential applications of wearable resonators. For instance, wearable self-powered sensors.

4.- This manuscript could incorporate a section on the main challenges of flexible resonators used as antenna-based systems, sensors and filters.

5.- The conclusions can be enhanced based on the above comments.

6.- The format of references must be revised. This format must be corrected.

Comments on the Quality of English Language

The English grammar is good.

Author Response

This manuscript presents a review on multiple approaches for wearable resonators used as antenna-based systems, sensors and filters. This review includes advantages and drawbacks of the flexible resonators, materials, fabrication processes, and applications. This manuscript can improve based on the following comments:

  1. This manuscript can add a section about modeling or numerical simulations (e.g., finite element models) on the performance of flexible resonators used as antenna-based systems, sensors and filters.

We appreciate the reviewer’s remark. However, the first focus of this paper is to perform a State-of-the-Art review of wearable microwave resonators for sensor design. The extension of the article might be a bit concerning if we include a section dedicated to modelling and numerical simulations. Nevertheless, we agree it is a crucial point and we have mentioned CST software and cited multiple articles, including figures, to expose the simulation procedure:

  • Section 4.2. ‘Ring Resonators in Flexible Electronics: Design Considerations’, sub-section ‘Bending’, reference [107] and Figure 9, page 25 of 48.
  • Section 5.3.3. ‘Evolution of the Embroidery Technique in the Design of Resonators’, reference [148], page 34 of 48.
  • Section 5.3.3. ‘Evolution of the Embroidery Technique in the Design of Resonators’, reference [17], page 34 of 48.
  • Section 5.3.3. ‘Evolution of the Embroidery Technique in the Design of Resonators’, reference [149], page 35 of 48.
  • Section 5.4. ‘Challenges in Implementing Resonators for Flexible Electronics Design’, sub-section ‘Performance, Materials and Dimensions’, reference [151], page 36 of 48.

  1. The authors can improve their manuscript by considering a section or information on the reliability tests of flexible resonators.

A paragraph about Reliability Tests is added at Section 5.1.1. under sub-section ‘SAR and Bending Reliability Tests of Flexible Resonator-based Sensors’, page 30 of 48.

  1. The authors can add more applications or potential applications of wearable resonators. For instance, wearable self-powered sensors.

We appreciate the reviewer’s remark. However, the scope of this article is passive electronics applications. The same is mentioned in Section 5.4. ‘Challenges in Implementing Resonators for Flexible Electronics Design’, sub-section ‘Power Transfer in Battery-Powered Devices’. We expose the challenge of integrating battery-powered devices for wearable applications and provide a brief explanation of it, however we may not endorse extending the scope of the article but considering it in a future investigation.

  1. This manuscript could incorporate a section on the main challenges of flexible resonators used as antenna-based systems, sensors and filters.

Section 5.4.’ Challenges in Implementing Resonators for Flexible Electronics Design’ is enhanced.

  1. The conclusions can be enhanced based on the above comments.

Conclusions are enhanced to include a reference to the new sections. Page 37 of 48.

  1. The format of references must be revised. This format must be corrected.

References have been revised and corrected.

Round 2

Reviewer 1 Report

Comments and Suggestions for Authors

The authors have improved their work. It Can be accepted for publication.

Comments on the Quality of English Language

Minor edits are needed.

Reviewer 2 Report

Comments and Suggestions for Authors

The authors improved their second version of manuscript considering the reviewer's comments.

Comments on the Quality of English Language

The quality of English language is good.